# TempFlow-GRPO: When Timing Matters for GRPO in Flow Models

**Xiaoxuan He**[1,2]*, **Siming Fu**[1]*, **Yuke Zhao**[1]*, **Wanli Li**[1], **Jian Yang**[2],
**Dacheng Yin**[2†‡], **Fengyun Rao**[2], **Bo Zhang**[1‡]

[1] ZheJiang University,
[2] WeChat Vision, Tencent Inc

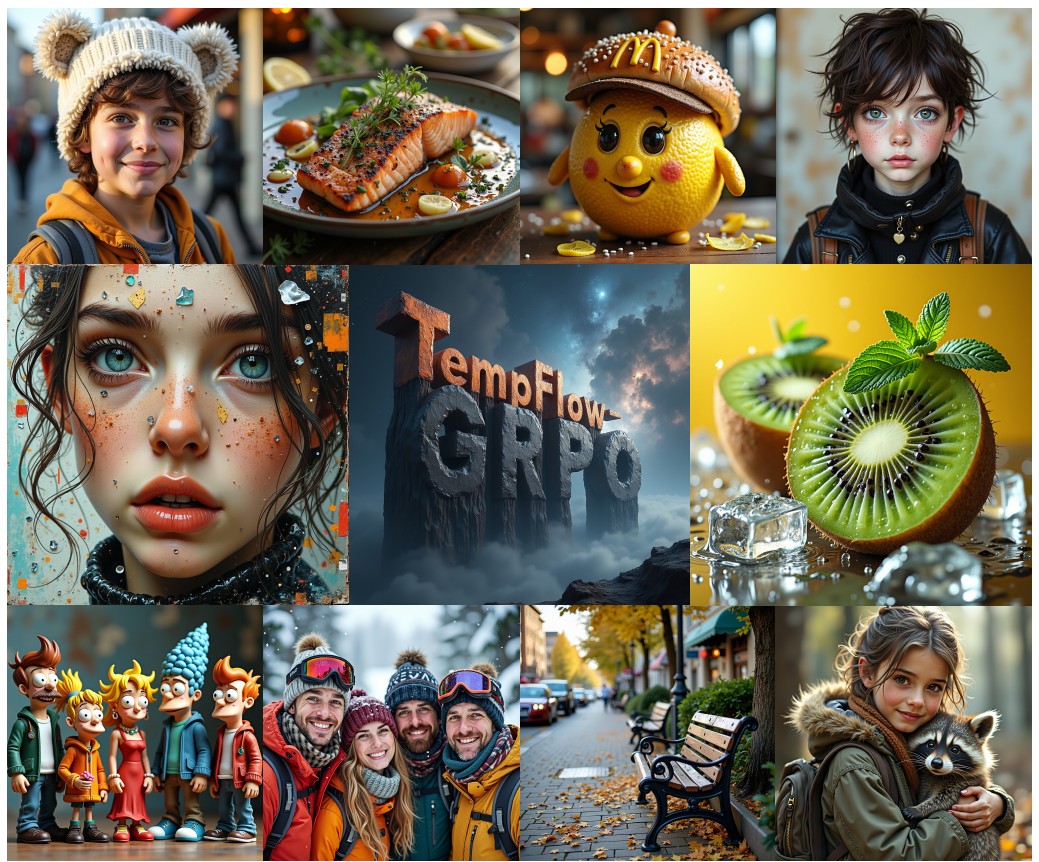

Figure 1: Images generated by our proposed TempFlow-GRPO with FLUX.1-dev. It substantially improves the baseline models, achieving superior photorealism and enhanced fine-grained detail.

## Abstract

Recent flow matching models for text-to-image generation have achieved remarkable quality, yet their integration with reinforcement learning for human preference alignment remains suboptimal, hindering fine-grained reward-based optimization. We observe that the key impediment to effective GRPO training of flow models is the temporal uniformity assumption in existing approaches: sparse terminal rewards with uniform credit assignment fail to capture the varying criticality of decisions across generation timesteps, resulting in inefficient exploration and suboptimal convergence. To remedy this shortcoming, we introduce **TempFlow-GRPO** (Temporal Flow-GRPO), a principled GRPO framework that captures and

---

* Equal Contribution.
† Project Leader.
‡ Corresponding authors.

exploits the temporal structure inherent in flow-based generation. TempFlow-GRPO introduces three key innovations: (i) a trajectory branching mechanism that provides process rewards by concentrating stochasticity at designated branching points, enabling precise credit assignment without requiring specialized intermediate reward models; (ii) a noise-aware weighting scheme that modulates policy optimization according to the intrinsic exploration potential of each timestep, prioritizing learning during high-impact early stages while ensuring stable refinement in later phases; and (iii) a seed group strategy that controls for initialization effects to isolate exploration contributions. These innovations endow the model with temporally-aware optimization that respects the underlying generative dynamics, leading to state-of-the-art performance in human preference alignment and text-to-image benchmarks.

# 1 INTRODUCTION

While text-to-image diffusion models have achieved unprecedented visual quality and semantic control (Esser et al., 2024; Xie et al., 2024a; Labs, 2024), aligning their outputs with human preference remains a formidable challenge. Reinforcement learning (Shao et al., 2024; Wu et al., 2025b) has emerged as a promising solution, giving rise to the field of Diffusion RL (Wallace et al., 2024; Black et al., 2023; Fan et al., 2023). However, the performance of these methods remains suboptimal, hindered by two fundamental limitations that have been largely overlooked: ***ignoring the temporal dynamics of generation*** and ***lacking intermediate feedback signals***. These approaches apply uniform optimization across all timesteps and provide rewards only at completion, missing the varying importance of decisions throughout the generation process.

The majority of existing approaches (Gu et al., 2024; Hong et al., 2024), including recent works like Flow-GRPO (Liu et al., 2025) and DanceGRPO (Xue et al., 2025), treat the multi-step generation process as a "black box" with temporally agnostic optimization. They apply uniform updates across all timesteps despite the fact that each timestep operates under different noise conditions and contributes differently to final image quality. Specifically, we plot the Figure 2 (left) with applying SDE at only one timestep in the entire ODE trajectory, which ensures that any deviation in the final reward can be attributed to the stochastic exploration introduced at that specific step. As shown in Figure 2 (left), the std of the reward varies dramatically between timesteps, reaching a peak during early structural decisions (steps 0-2) and approaching zero during final refinements (steps 6-8). Yet Flow-GRPO maintains uniform treatment throughout, squandering high-impact exploration opportunities. Alternative approaches like SPO (Liang et al., 2024) attempt to address temporal dynamics through process reward models, but training such models on semantically ambiguous intermediate states is notoriously difficult. This raises a fundamental question: ***how can we effectively achieve precise credit assignment for intermediate actions while adapting optimization intensity to each timestep's exploration capacity?***

We address these limitations with **TempFlow-GRPO**, a temporally-aware RL framework built on two key insights. **First**, we define the visualization method in the left of Figure 2 as trajectory branching, which enables precise credit assignment by strategically introducing stochasticity at individual timesteps while maintaining deterministic evolution elsewhere (Figure 2, Right). This provides *provable* guarantees: (1) reward variance localizes to the branching point, (2) improvements are directly attributable to specific exploration outcomes, and (3) existing reward models require no modification. **Second**, noise-aware policy weighting modulates optimization intensity based on each timestep's intrinsic noise level. Early high-noise stages receive larger weight updates to encourage structural exploration, while late low-noise stages receive gentler updates to preserve learned features. **Third**, we introduce a seed-level grouping strategy that controls for initial noise influence, ensuring reward variations are attributed to branching exploration rather than random initialization. Together, these mechanisms create a framework that is conceptually simple, computationally efficient, and seamlessly integrates into existing flow matching architectures—all while respecting the temporal dynamics that uniform approaches ignore.

Our main contributions are threefold:

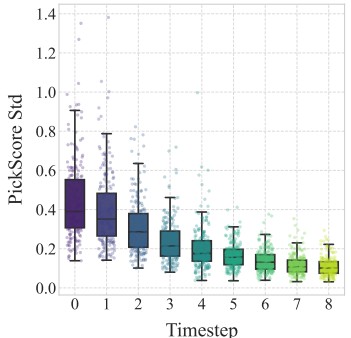 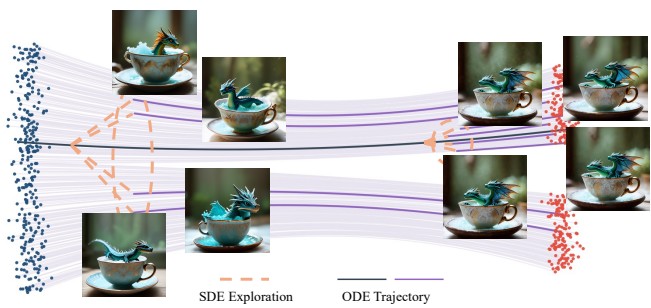

Figure 2: (Left) Reward Variance Analysis: We plot the standard deviation of PickScore at each denoising step for 200 prompts, per prompt group size is 24. The results, obtained via applying SDE at only one step, reveal that reward variance is highest in the initial steps, indicating that early-stage interventions are most impactful for exploration. (Right) Method Illustration: By branching a stochastic (SDE) exploration from a specific, known state on a deterministic (ODE) trajectory, the resulting difference in the final reward can be unambiguously attributed to the exploration action taken at that precise branching point.

- We pinpoint temporal uniformity—the equal treatment of all timesteps—as the primary limitation of flow-based GRPO. Our proposed **TempFlow-GRPO** overcomes this by introducing two key innovations: precise credit assignment to intermediate actions and noise-aware adaptation of optimization intensity.

- We introduce **trajectory branching** and **noise-aware reweighting** mechanisms to learn temporally-structured policies that respect the inherent dynamics of generative models. Additionally, we propose an efficient **seed group strategy** that effectively isolates exploration effects and considerably enhances overall performance.

- We demonstrate state-of-the-art performance on standard text-to-image benchmarks, achieving superior sample quality, human preference alignment, and compositional image generation compared to existing flow-based RL methods.

## 2 RELATED WORK

**Alignment for Diffusion Models.** Alignment for diffusion models has emerged as a key research area. D3PO (Yang et al., 2024) introduces Direct Preference for Denoising Diffusion Policy Optimization to directly fine-tune diffusion models. Diffusion-DPO (Wallace et al., 2024) adapts Direct Preference Optimization (DPO (Rafailov et al., 2023)) as a simpler RLHF alternative that directly optimizes policies satisfying human preferences. DyMO (Xie & Gong, 2025) proposes a training-free alignment method for inference-time preference alignment. Flow-GRPO (Liu et al., 2025) first integrates online reinforcement learning into flow matching models. However, Flow-GRPO applies uniform optimization across timesteps and suffers from sparse terminal rewards, ignoring time-varying exploration potential in stochastic processes. Our TempFlow-GRPO addresses these limitations through trajectory branching for precise credit assignment and noise-aware policy weighting aligned with natural exploration capacity.

**Process Reward.** Shaping reward processes beyond sparse terminal rewards significantly improves learning and policy performance. Zhang et al. (Zhang et al., 2025) combines response-level and step-level evaluation metrics. ThinkPRM (Khalifa et al., 2025) builds data-efficient process reward models (PRMs) using verification chain-of-thought for step-wise verification. In diffusion models, SPO (Liang et al., 2024) trains step-aware preference models for both noisy and clean images. However, PRMs require expensive step-level supervision. Methods like PRIME (Cui et al., 2025) enable online PRM updates using only outcome labels through implicit process rewards. Math-Shepherd (Wang et al., 2023) assigns reward scores to solution steps. Given challenges in scoring noisy images, algorithms enabling process rewards for flow models are needed. Our TempFlow-

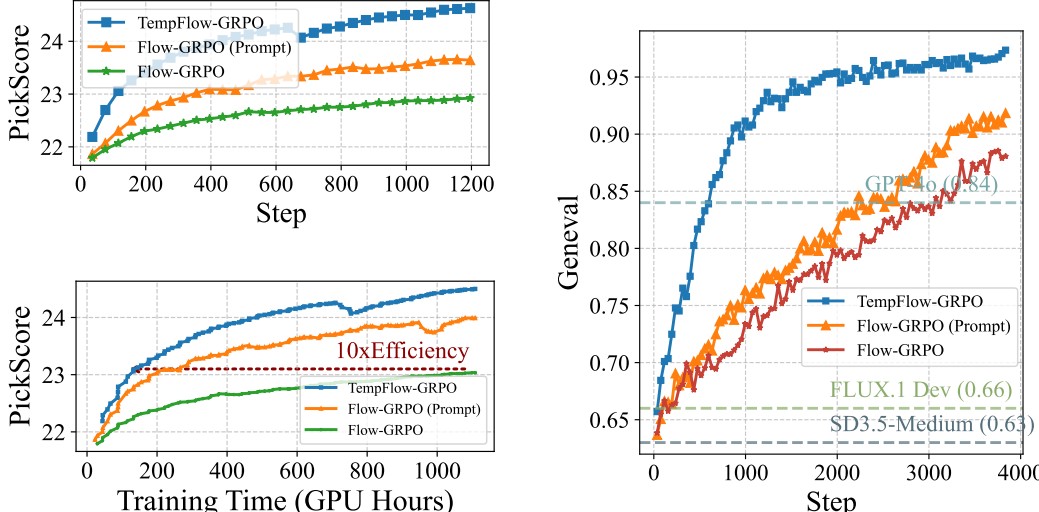

Figure 3: (Left) Performance comparison on the PickScore benchmark across training steps and GPU hours. Flow-GRPO (Prompt) represents **an improved baseline with group-wise standard deviation stabilization**. TempFlow-GRPO consistently outperforms both Flow-GRPO variants in both sample efficiency (steps) and computational efficiency (GPU hours), **demonstrating superior training efficiency while achieving the best performance**. (Right) On the Geneval benchmark, TempFlow-GRPO achieves the highest performance, significantly outperforming Flow-GRPO and surpassing state-of-the-art models including GPT-4o, FLUX.1 Dev, and SD3.5-Medium.

GRPO circumvents specialized process reward models by directly attributing outcome-based signals to intermediate actions, enabling precise credit assignment without computational overhead of training step-level evaluators for semantically ambiguous states.

## 3 PRELIMINARY: FLOW-GRPO

**GRPO.** RL aims to learn a policy that maximizes the expected cumulative reward. GRPO optimizes the policy model by maximizing the following objective:

$$\mathcal{J}_{\text{Flow-GRPO}}(\theta) = \mathbb{E}_{\boldsymbol{c}\sim\mathcal{C},\{\boldsymbol{x}^i\}_{i=1}^G\sim\pi_{\theta_{\text{old}}}(\cdot|\boldsymbol{c})} f(r, \hat{A}, \theta, \epsilon, \beta) \tag{1}$$

$$f(r, \hat{A}, \theta, \epsilon, \beta) = \frac{1}{G}\sum_{i=1}^G \frac{1}{T}\sum_{t=0}^{T-1}(min(r_t^i(\theta)\hat{A}_t^i, \text{clip}(r_t^i(\theta), 1-\epsilon, 1+\epsilon)\hat{A}_t^i) - \beta D_{KL}(\pi_\theta||\pi_{\text{ref}}))$$

$$r_t^i(\theta) = \frac{p_\theta(\boldsymbol{x}_{t-1}^i|\boldsymbol{x}_t^i, \boldsymbol{c})}{p_{\theta_{\text{old}}}(\boldsymbol{x}_{t-1}^i|\boldsymbol{x}_t^i, \boldsymbol{c})}, T \text{ is the timestep.} \tag{2}$$

Given a prompt $\boldsymbol{c}$, the flow model $p_\theta$ samples a group of $G$ individual images $\{\boldsymbol{x}_0^i\}_{i=1}^G$ and the corresponding reverse-time trajectories $\{(\boldsymbol{x}_T^i, \boldsymbol{x}_{T-1}^i, ..., \boldsymbol{x}_0^i)\}_{i=1}^G$. Then, the advantage of the $i$-th image is calculated by normalizing the group-level rewards as follows:

$$\hat{A}_t^i = \frac{R(\boldsymbol{x}_0^i, \boldsymbol{c}) - \text{mean}(\{R(\boldsymbol{x}_0^i, \boldsymbol{c})\}_{i=1}^G)}{\text{std}(\{R(\boldsymbol{x}_0^i, \boldsymbol{c})\}_{i=1}^G)} \tag{3}$$

**Convert ODE to SDE.** Flow-GRPO converts the deterministic ODE into an equivalent SDE that matches the original model's marginal probability density function at all timesteps. The ODE and SDE is as follows:

$$d\boldsymbol{x}_t = \boldsymbol{v}_t dt \tag{4}$$

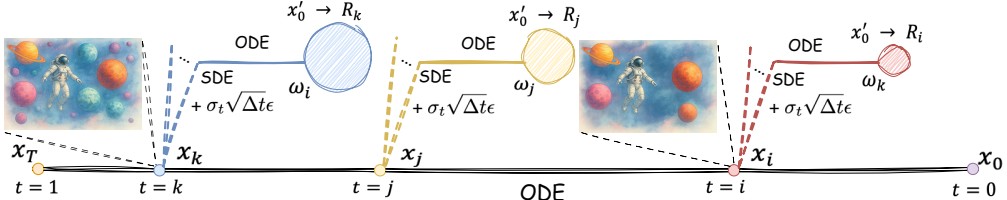

Figure 4: **Overview of TempFlow-GRPO Framework.** Our method performs trajectory branching by switching from ODE to SDE sampling at selected timesteps (t=k, j, i), injecting noise $\sigma_t\sqrt{\Delta t}\epsilon$ to create exploratory branches. Each branch generates a distinct outcome with reward $R_i$, enabling precise credit assignment. The framework applies noise-aware weighting where $\omega_i > \omega_j > \omega_k$, prioritizing optimization at high-noise early stages (larger circles) over low-noise refinement phases (smaller circles), aligning learning intensity with each timestep's intrinsic exploration capacity. We visualize the model's learning process as an astronaut exploring unknown planets: in early stages , the model explores vast possibility spaces with high uncertainty, while later stages involve focused navigation toward the final destination.

$$\boldsymbol{x}_{t+\Delta t} = \boldsymbol{x}_t + [\boldsymbol{v}_\theta(\boldsymbol{x}_t, t) + \frac{\sigma_t^2}{2t}(\boldsymbol{x}_t + (1-t)\boldsymbol{v}_\theta(\boldsymbol{x}_t, t))]\Delta t + \sigma_t\sqrt{\Delta t}\epsilon \qquad (5)$$

where $\epsilon \sim \mathcal{N}(0, \boldsymbol{I})$ injects stochasticity and $\sigma_t = a\sqrt{\frac{t}{1-t}}$. And the KL divergence between $\pi_\theta$ and the reference policy $\pi_{\text{ref}}$ is a closed form:

$$D_{KL}(\pi_\theta||\pi_{\text{ref}}) = \frac{||\bar{\boldsymbol{x}}_{t+\Delta t,\theta} - \bar{\boldsymbol{x}}_{t+\Delta t,\text{ref}}||}{2\sigma_t^2\Delta t} = \frac{\Delta t}{2}(\frac{\sigma_t(1-t)}{2t} + \frac{1}{\sigma_t})^2||\boldsymbol{v}_\theta(\boldsymbol{x}_t, t) - \boldsymbol{v}_{\text{ref}}(\boldsymbol{x}_t, t)||^2 \quad (6)$$

## 4 METHODS

### 4.1 TEMPORAL FLOW-GRPO

Flow-GRPO advances online RL for flow matching but overlooks time-dependent generative dynamics. We identify two key limitations: **(1) Sparse Terminal Rewards**: uniform credit assignment across timesteps fails to distinguish critical early decisions from later fine-tuning; **(2) Uniform Optimization**: ignoring non-uniform noise in SDE sampling, where high-noise early steps offer greater exploration potential than low-noise refinement phases (Xie & Gong, 2025). To address these issues, we propose **TempFlow-GRPO** (Figure 4), introducing process rewards via trajectory branching and noise-aware policy weighting.

#### 4.1.1 TRAJECTORY BRANCHING FOR PROCESS REWARDS

Traditional process reward methods require specialized reward models to evaluate noisy intermediate states $\boldsymbol{x}_t$, which is exceptionally difficult due to the semantic ambiguity of partially-denoised representations. We propose an elegant alternative that leverages the deterministic-stochastic sampling methods of flow matching models.

**Key Insight**: Instead of training complex process reward models, we use existing high-quality outcome-based reward models and directly attribute their scores to intermediate exploratory actions through a novel trajectory branching mechanism.

**Proposition (Trajectory Branching)**: We define a trajectory branching operation where a trajectory evolves deterministically until a **designated branching timestep** $k$, where $\boldsymbol{x}_k$ is obtained from initial noise $\boldsymbol{x}_T$ using Equation 4. At this branching point, stochasticity is introduced via noise variable $\epsilon$ in Equation 5, yielding $\boldsymbol{x}_{k-1} = \text{SDE}(\boldsymbol{x}_k, \epsilon)$. The remainder of the trajectory $\boldsymbol{x}_{k-2}, \boldsymbol{x}_{k-3}, \ldots, \boldsymbol{x}_0$ is generated deterministically as $\boldsymbol{x}_0 = \text{ODE}^{k-1}(\boldsymbol{x}_{k-1})$, $\text{ODE}^{k-1}$ denotes $k-1$ times ODE.

**Theorem (Credit Localization)**: Since all stochasticity and model controllability are concentrated at the branching point, the total reward variance and all parameter-dependent improvements are entirely attributable to the outcome of noise injection at $k$. This enables rigorous and efficient credit

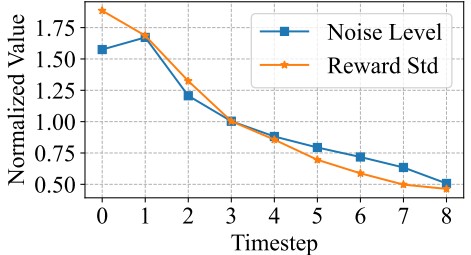 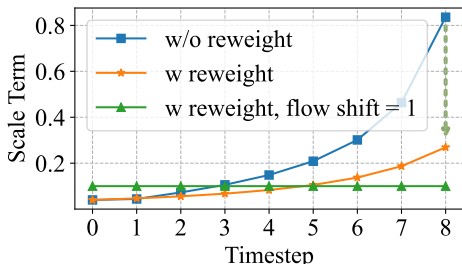

Figure 5: **(Left)** Strong correlation between reward standard deviation and noise level across generative timesteps. **(Right)** Scale term analysis reveals a fundamental mismatch in standard GRPO: scale terms are inversely proportional to noise levels, causing low-noise refinement steps to dominate optimization despite minimal impact on image content.

assignment localized to the branching point.

In practice, we replace the reward for the $k$-th step from $R(\boldsymbol{x}_0^i, \boldsymbol{c})$ to $R(\text{ODE}^{k-1}(\text{SDE}(\boldsymbol{x}_k^i, \boldsymbol{\epsilon}^i)), \boldsymbol{c})$, where $\boldsymbol{x}_0^i$ is sampled with SDE and $\text{ODE}^{k-1}(\text{SDE}(\boldsymbol{x}_k^i, \boldsymbol{\epsilon}^i))$ is sampled with ODE-SDE-ODE. Trajectory branching allows for the precise attribution of the terminal reward to step $k$, effectively creating a temporally-aware reward signal.

### 4.1.2 NOISE-AWARE POLICY WEIGHTING

While trajectory branching provides precise reward signals for individual timesteps, the generative process consists of a sequence of $T$ potential branching points with fundamentally different characteristics. The SDE sampler exhibits time-varying stochasticity: the noise injection magnitude $\sigma_t\sqrt{\Delta t}$ is large during initial generation stages and diminishes to near zero during final refinement stages. This non-uniform noise distribution implies that exploration capacity varies dramatically across timesteps. An exploratory action at an early stage has vastly different impact and risk compared to perturbations on near-perfect images. However, standard GRPO applies uniform optimization pressure, implicitly assuming equal learning importance at all stages.

**Reweighting by Noise Level.** We wonder if the level of the exploration space itself can serve as a proxy for a reweighting factor. We visualized the noise level alongside the reward standard deviation (Figure 5, left) and observed a striking correspondence between the two. This strong correlation suggests that the noise level, serves as an excellent and intrinsic proxy for the exploration capacity and associated risk at each timestep.

We therefore propose to reweight the policy loss directly using the noise level. For each timestep $t$, we introduce a weighting factor proportional to the noise level. Specifically, we modify the original policy loss function to the following weighted form:

$$\mathcal{J}_{\text{policy}}(\theta) = \frac{1}{G}\sum_{i=1}^{G}\frac{1}{T}\sum_{t=0}^{T-1}\text{Norm}(\sigma_t\sqrt{\Delta t})(min(r_t^i(\theta)\hat{A}_t^i, \text{clip}(r_t^i(\theta), 1-\epsilon, 1+\epsilon)\hat{A}_t^i) \quad (7)$$

The intuition behind this weighting strategy is to align the optimization pressure with the inherent properties of the generative process. In the early stages of generation, noise is large, amplifying the learning signal during these high-noise, high-impact phases and encouraging the model to perform effective macroscopic exploration. As generation proceeds, noise diminishes, shifts the optimization focus towards fine-grained adjustments and stability, preventing aggressive exploration from corrupting a high-fidelity state with noise or artifacts.

### 4.2 POLICY GRADIENT-BASED THEORETICAL JUSTIFICATION

Consider a generative process parameterized by $\theta$, the policy gradient can be written as:

$$\nabla_\theta\mathcal{J}(\theta) = \sum_{k=0}^{T-1}\mathbb{E}_{\boldsymbol{x}_T\sim\mathcal{N}(0,\boldsymbol{I}), \boldsymbol{\epsilon}\sim\mathcal{N}(0,\boldsymbol{I})}[\nabla_\theta\log p_\theta(\boldsymbol{x}_{k-1}|\boldsymbol{x}_k)\hat{A}_k] \quad (8)$$

This yields:

$$\nabla_\theta \mathcal{J}(\theta) = \sum_{k=0}^{T-1} \mathbb{E}_{\boldsymbol{x}_T \sim \mathcal{N}(0,\boldsymbol{I}), \boldsymbol{\epsilon} \sim \mathcal{N}(0,\boldsymbol{I})} \left[ \left(\frac{1}{a} + \frac{a}{2}\right) \underbrace{\sqrt{\frac{\Delta k(1-k)}{k}}}_{\text{Scale Term}} \cdot \boldsymbol{\epsilon} \cdot \nabla_\theta \boldsymbol{v}_\theta(\boldsymbol{x}_k, k) \hat{A}_k \right] \quad (9)$$

This reveals that the natural gradient coefficient is proportional to $\sqrt{\frac{\Delta k(1-k)}{k}}$, which captures the intrinsic exploration potential at timestep $k$. After reweighting, we have the following derivation:

$$\nabla_\theta \mathcal{J}(\theta) = \sum_{k=0}^{T-1} \mathbb{E}_{\boldsymbol{x}_T \sim \mathcal{N}(0,\boldsymbol{I}), \boldsymbol{\epsilon} \sim \mathcal{N}(0,\boldsymbol{I})} \left[ \left(\frac{1}{a} + \frac{a}{2}\right) \underbrace{\Delta k}_{\text{Scale Term}} \cdot \boldsymbol{\epsilon} \cdot \nabla_\theta \boldsymbol{v}_\theta(\boldsymbol{x}_k, k) \hat{A}_k \right] \quad (10)$$

where $\mathbb{E}_{\boldsymbol{\epsilon}}[\boldsymbol{\epsilon} \hat{A}_k] = \frac{g_k}{||g_k||}$, indicating that the norm of $\mathbb{E}_{\boldsymbol{\epsilon}}[\boldsymbol{\epsilon} \hat{A}_k]$ is invariant among the timesteps. Additional details about these equations are provided in Appendix A.1. In Equation 9 and Equation 10, the scale terms that modulate how each timestep's model gradient $\nabla_\theta \boldsymbol{v}_\theta(\boldsymbol{x}_k, k)$ contributes to the overall reward gradient simplify to distinct functions, which we denote as $\sqrt{\frac{\Delta k(1-k)}{k}}$ and $\Delta k$. We visualize these scale terms under different flow shifts in Figure 5 (right). The "w/o reweighting" setting exhibits highly imbalanced gradient contributions across timesteps, where early denoising steps performing broad structural exploration contribute significantly less than late steps focused on fine-grained refinement. Our noise-aware policy reweighting mitigates this issue by simplifying the scale term to be proportional to step size $\Delta k$. When flow shift equals 1, our method achieves perfect equilibrium with equal gradient contributions from all timesteps, completely balancing the effect across the generation trajectory. **Additional details about this section are provided in Appendix A.1**.

## 4.3 GROUP STRATEGY

**Batch Group & Prompt Group.** The original GRPO (Shao et al., 2024) framework employs a group-level strategy, where trajectories are grouped based on a shared prompt. To mitigate issues of overfitting and instability, Reinforce++ (Hu et al., 2025) later introduced a batch-level normalization.

**Seed Group.** In our work, trajectory branching requires $K$ distinct explorations at each timestep. Furthermore, we propose a novel seed-level grouping strategy. Under this approach, trajectories originating from the same prompt are further grouped if they share an identical initial noise. This methodology effectively controls for the influence of the initial noise, thereby ensuring that variations in reward can be attributed solely to the exploration conducted during the branching process. The experimental results, presented in Figure 6, validate our approach. As shown, TempFlow-GRPO consistently achieves superior performance compared to Flow-GRPO, regardless of the grouping strategy employed.

## 5 EXPERIMENT

Following Flow-GRPO, we validate our approach on the Compositional Image Generation benchmark in Geneval (Ghosh et al., 2023) and the Human Preference Alignment task in PickScore (Kirstain et al., 2023). To ensure a fair comparison, our experimental setup is kept consistent with that of Flow-GRPO. For instance, we normalized the weights applied to the policy loss to have a mean of 1 at all timesteps. While our method introduces a trajectory branching strategy, we maintain an identical group configuration by setting 4 initial noise seeds and a branching factor ($K$) of 6. This results in a total group size of 24 (4 seeds × 6 branches) and 48 groups, perfectly matching the setup in Flow-GRPO. Furthermore, to benchmark our method against concurrent work, we conduct additional experiments on the FLUX.1-dev model. **More detailed experimental settings (include comparsion with DanceGRPO) can be found in the Appendix A.2.**

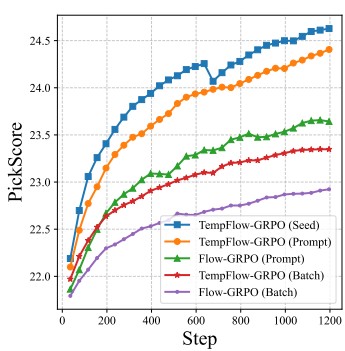

Figure 6: Performance of TempFlow-GRPO and Flow-GRPO on different group strategy.

Table 1: **GenEval Result.** Best scores are in blue , second-best in green . Results for models are from Flow-GRPO. Obj.: Object; Attr.: Attribution.

| Model | Step | Overall ↑ | Single Obj. ↑ | Two Obj. ↑ | Counting ↑ | Colors ↑ | Position ↑ | Attr. Binding ↑ |
|---|---|---|---|---|---|---|---|---|
| *Diffusion Models* | | | | | | | | |
| LDM (Rombach et al., 2022) | - | 0.37 | 0.92 | 0.29 | 0.23 | 0.70 | 0.02 | 0.05 |
| SD1.5 (Rombach et al., 2022) | - | 0.43 | 0.97 | 0.38 | 0.35 | 0.76 | 0.04 | 0.06 |
| SD2.1 (Rombach et al., 2022) | - | 0.50 | 0.98 | 0.51 | 0.44 | 0.85 | 0.07 | 0.17 |
| SD-XL (Podell et al., 2023) | - | 0.55 | 0.98 | 0.74 | 0.39 | 0.85 | 0.15 | 0.23 |
| DALLE-2 (Ramesh et al., 2022) | - | 0.52 | 0.94 | 0.66 | 0.49 | 0.77 | 0.10 | 0.19 |
| DALLE-3 (Betker et al., 2023) | - | 0.67 | 0.96 | 0.87 | 0.47 | 0.83 | 0.43 | 0.45 |
| *Autoregressive Models* | | | | | | | | |
| Show-o (Xie et al., 2024b) | - | 0.53 | 0.95 | 0.52 | 0.49 | 0.82 | 0.11 | 0.28 |
| Emu3-Gen (Wang et al., 2024) | - | 0.54 | 0.98 | 0.71 | 0.34 | 0.81 | 0.17 | 0.21 |
| JanusFlow (Ma et al., 2025a) | - | 0.63 | 0.97 | 0.59 | 0.45 | 0.83 | 0.53 | 0.42 |
| Janus-Pro-7B (Chen et al., 2025) | - | 0.80 | 0.99 | 0.89 | 0.59 | 0.90 | 0.79 | 0.66 |
| GPT-4o (Hurst et al., 2024) | - | 0.84 | 0.99 | 0.92 | 0.85 | 0.92 | 0.75 | 0.61 |
| *Flow Matching Models* | | | | | | | | |
| FLUX.1 Dev (Labs, 2024) | - | 0.66 | 0.98 | 0.81 | 0.74 | 0.79 | 0.22 | 0.45 |
| SD3.5-L (Esser et al., 2024) | - | 0.71 | 0.98 | 0.89 | 0.73 | 0.83 | 0.34 | 0.47 |
| SANA-1.5 4.8B (Xie et al., 2025) | - | 0.81 | 0.99 | 0.93 | 0.86 | 0.84 | 0.59 | 0.65 |
| SD3.5-M (Esser et al., 2024) | - | 0.63 | 0.98 | 0.78 | 0.50 | 0.81 | 0.24 | 0.52 |
| *GRPO based Methods* | | | | | | | | |
| SD3.5-M+Flow-GRPO (Liu et al., 2025) | 5600 | 0.95 | 1.00 | 0.99 | 0.95 | 0.92 | 0.99 | 0.86 |
| SD3.5-M+Flow-GRPO (Liu et al., 2025) | 3800 | 0.88 | 0.99 | 0.96 | 0.90 | 0.87 | 0.83 | 0.78 |
| **SD3.5-M+TempFlow-GRPO** | 3800 | 0.97 | 1.00 | 1.00 | 0.96 | 0.95 | 0.99 | 0.91 |

## 5.1 MAIN RESULTS

**Compositional Image Generation.** We evaluate the compositional image generation capability of TempFlow-GRPO on the Geneval benchmark with its corresponding reward model. The experimental results are summarized in Table 1. As shown, our approach significantly improves the performance of the base model, increasing the overall score from 0.63 to 0.97. Furthermore, among GRPO-based methods, our method substantially outperforms Flow-GRPO: it achieves a performance of 0.97 within only 3,800 steps, whereas Flow-GRPO reaches only 0.88 under the same conditions. Additionally, as illustrated in Figure 3, our method requires only about 2,000 steps to achieve a score of 0.95, while Flow-GRPO needs approximately 5,600 steps to reach the same level. Overall, these results demonstrate that TempFlow-GRPO not only accelerates convergence but also achieves superior final performance compared to existing approaches.

**Human Preference Alignment.** To further validate the generalizability of our approach, we conducted experiments on the PickScore benchmark, using PickScore as the reward model. As shown in Figure 3 (left), TempFlow-GRPO, achieves the highest performance, surpassing the original Flow-GRPO by approximately 1.7% and outperforming the improved baseline Flow-GRPO (Prompt) by about 1.0%. Notably, our method requires only 100–200 training steps to match the performance of Flow-GRPO, and just 300–400 steps to reach the level of Flow-GRPO (Prompt). These results on PickScore further demonstrate the general applicability of our method as a unified flow-based RL algorithm across different reward models.

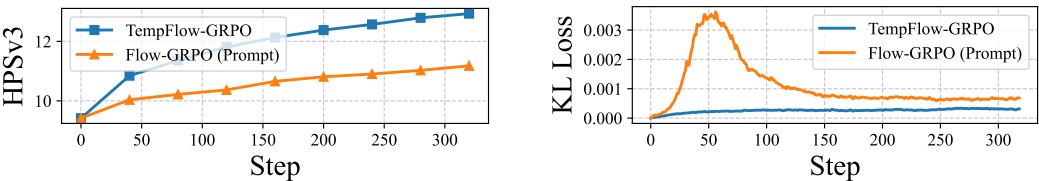

Figure 7: The performance and KL loss (smoothed) of HPSv3 on PickScore dataset (FLUX.1-dev).

**HPSv3 on FLUX.1-dev.** To further validate the performance of TempFlow-GRPO across different model and reward models, we select FLUX.1-dev as our base model with 1024 resolution and employ HPSv3 (Ma et al., 2025b) as the reward model. As shown in Figure 7, which presents the results, our method requires only 80 steps to match the performance that Flow-GPO achieves in 300

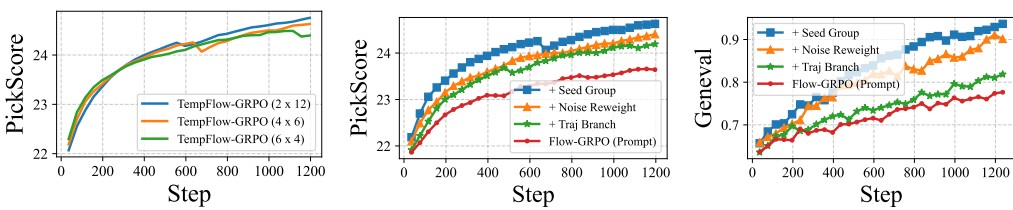

Figure 8: Ablation studies on trajectory branch and noise-aware policy reweighting.

steps. Moreover, our approach maintains a lower and more stable KL loss. **Further experimental results and visualizations for the FLUX model can be found in the Appendix. A.3.**

## 5.2 ANALYSIS

**Ablation of Trajectory branching.** In our trajectory branching strategy, branching is performed at each step. To ensure a fair comparison with Flow-GRPO, we maintain a constant total group size of 24. Therefore, we tested several branching configurations: 2 initial noises with 12 branches each (2×12), 4 initial noises with 6 branches each (4×6), and 6 initial noises with 4 branches each (6×4). The experimental results are shown in the left of Figure 8. The figure indicates that a larger number of initial noises helps to accelerate performance improvement in the early stages of training. However, as training progresses, a higher number of branches yields better results. To strike a balance, we ultimately selected the 4×6 configuration as the default setting for our paper.

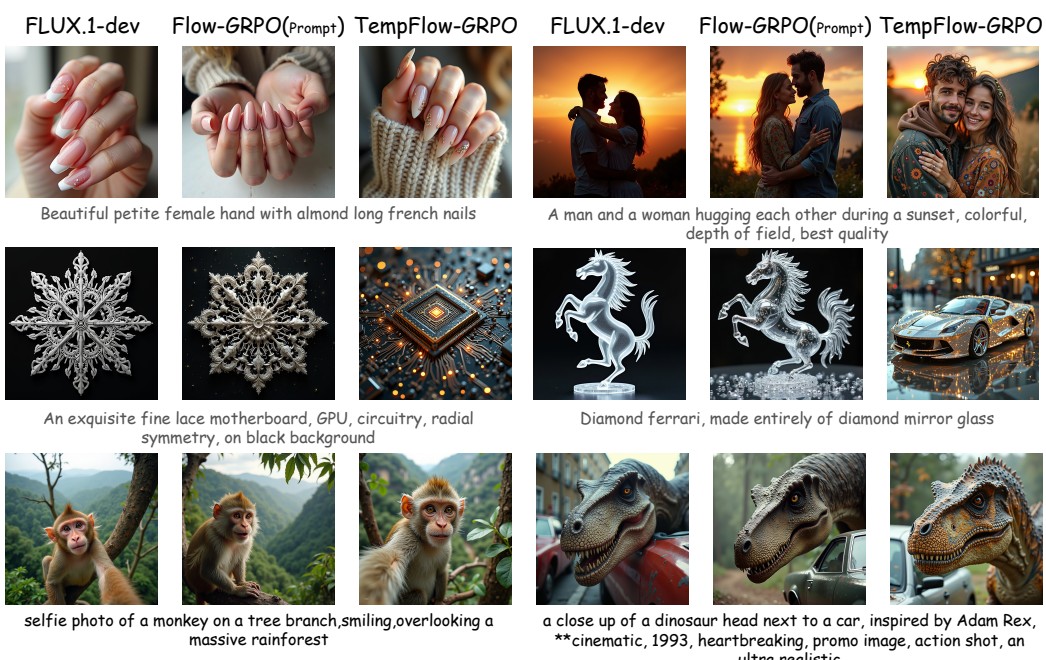

Figure 9: Qualitative comparison between FLUX.1-dev, Flow-GRPO (Prompt) and TempFlow-GRPO with HPSv3 as reward on PickScore prompts.

**Ablation of TempFlow-GRPO.** We conducted ablation studies to explore the effectiveness of our proposed components: the trajectory branch, noise-aware policy weighting and seed group. These ablations were performed on both the Geneval and PickScore benchmarks. As shown in the second and third subfigures of Figure 8, on the PickScore benchmark, introducing the trajectory branch further improves the performance of Flow-GRPO (Prompt), and applying noise-aware reweighting on top of this further boosted the performance. Finally, we achieved the highest performance by

applying seed group. On the Geneval benchmark, the benefit of the noise-aware strategy is even more significant: compared to Flow-GRPO, noise-aware policy reweighting boosts performance from 0.82 to 0.92 in 1200 step, a 10% improvement while the trajectory branch also brings about a substantial gain of approximately 5% and seed group achieves 2% improvement. These ablation results clearly demonstrate the effectiveness of our proposed methods.

**Qualitative Result.** We also conducted qualitative analyses on the FLUX.1-dev, Flow-GRPO (Prompt), and TempFlow-GRPO. As shown in Figure 9, compared to Flow-GRPO (Prompt), TempFlow-GRPO produces images with noticeably finer details and fewer visual artifacts or mistakes. In particular, our approach demonstrates superior capability in preserving complex structures and realistic textures. These qualitative improvements further highlight the advantages of our method in generating high-quality, visually appealing images.

## 6 CONCLUSION

We presented TempFlow-GRPO, a temporally-aware reinforcement learning framework that addresses fundamental limitations in existing flow-based GRPO methods. Through trajectory branching, we enable precise credit assignment to intermediate actions without requiring specialized process reward models. Through noise-aware weighting, we ensure that optimization intensity matches each timestep's exploration potential. We also introduce a principled seed group strategy that controls for initialization effects. Our extensive experiments demonstrate that TempFlow-GRPO achieves SOTA performance on human preference.

**Limitations.** Although our method achieves significant improvements in both performance and image quality, the current experiments focus primarily on algorithmic innovations rather than reward model enhancements. **In future work, we plan to explore how to effectively incorporate multimodal rewards from more powerful foundation models and develop comprehensive reward frameworks, aiming to enhance performance across multiple evaluation dimensions.**

**Use of LLMs.** We utilize LLMs to assist with formula derivations and writing refinement.

## ACKNOWLEDGEMENTS

This work was partially supported by the National Natural Science Foundation of China under Grant No. 62402434.

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

# A APPENDIX

## A.1 POLICY GRADIENT-BASED THEORETICAL FRAMEWORK

To provide a deeper understanding of our approach, we now examine it from the policy gradient perspective. **Note that in the summation $\sum_{k=0}^{T-1}$, the index $k$ denotes the timestep, while in subsequent equations, $k$ represents the timestep value.** Simplifying Equation 5, we obtain $\boldsymbol{x}_{k-1} \sim \mathcal{N}(\mu_\theta(\boldsymbol{x}_k, k), \sigma_k^2 \Delta k \boldsymbol{I})$, where:

$$\mu_\theta(\boldsymbol{x}_k, k) = \boldsymbol{x}_k + \left[\boldsymbol{v}_\theta(\boldsymbol{x}_k, k) + \frac{\sigma_k^2}{2k}(\boldsymbol{x}_k + (1-k)\boldsymbol{v}_\theta(\boldsymbol{x}_k, k))\right]\Delta k \tag{11}$$

Starting from the policy gradient formulation in Equation 8, we have:

$$\nabla_\theta \mathcal{J}(\theta) = \sum_{k=0}^{T-1} \mathbb{E}_{\boldsymbol{x}_T \sim \mathcal{N}(0,\boldsymbol{I}), \boldsymbol{\epsilon} \sim \mathcal{N}(0,\boldsymbol{I})}[\nabla_\theta \log p_\theta(\boldsymbol{x}_{k-1}|\boldsymbol{x}_k)\hat{A}_k] \tag{12}$$

Substituting $\boldsymbol{x}_{k-1}$ in the log-probability:

$$\nabla_\theta \mathcal{J}(\theta) = \sum_{k=0}^{T-1} \mathbb{E}_{\boldsymbol{x}_T \sim \mathcal{N}(0,\boldsymbol{I}), \boldsymbol{\epsilon} \sim \mathcal{N}(0,\boldsymbol{I})}\left[\nabla_\theta \log \exp\left(-\frac{\|\boldsymbol{x}_{k-1} - \mu_\theta(\boldsymbol{x}_k, k)\|^2}{2\sigma_k^2 \Delta k}\right)\hat{A}_k\right] \tag{13}$$

$$= \sum_{k=0}^{T-1} \mathbb{E}_{\boldsymbol{x}_T \sim \mathcal{N}(0,\boldsymbol{I}), \boldsymbol{\epsilon} \sim \mathcal{N}(0,\boldsymbol{I})}\left[\nabla_\theta\left(-\frac{\|\boldsymbol{x}_{k-1} - \mu_\theta(\boldsymbol{x}_k, k)\|^2}{2\sigma_k^2 \Delta k}\right)\hat{A}_k\right] \tag{14}$$

Taking the gradient with respect to $\theta$:

$$\nabla_\theta \mathcal{J}(\theta) = \sum_{k=0}^{T-1} \mathbb{E}_{\boldsymbol{x}_T \sim \mathcal{N}(0,\boldsymbol{I}), \boldsymbol{\epsilon} \sim \mathcal{N}(0,\boldsymbol{I})}\left[\frac{\boldsymbol{x}_{k-1} - \mu_\theta(\boldsymbol{x}_k, k)}{\sigma_k^2 \Delta k} \cdot \nabla_\theta \mu_\theta(\boldsymbol{x}_k, k)\hat{A}_k\right] \tag{15}$$

Since $\boldsymbol{x}_{k-1} = \mu_\theta(\boldsymbol{x}_k, k) + \sigma_k\sqrt{\Delta k} \cdot \boldsymbol{\epsilon}$ where $\boldsymbol{\epsilon} \sim \mathcal{N}(0, \boldsymbol{I})$:

$$\nabla_\theta \mathcal{J}(\theta) = \sum_{k=0}^{T-1} \mathbb{E}_{\boldsymbol{x}_T \sim \mathcal{N}(0,\boldsymbol{I}), \boldsymbol{\epsilon} \sim \mathcal{N}(0,\boldsymbol{I})}\left[\frac{\boldsymbol{\epsilon}}{\sigma_k\sqrt{\Delta k}} \cdot \nabla_\theta \mu_\theta(\boldsymbol{x}_k, k)\hat{A}_k\right] \tag{16}$$

Expanding $\nabla_\theta \mu_\theta(\boldsymbol{x}_k, k)$:

$$\nabla_\theta \mu_\theta(\boldsymbol{x}_k, k) = \nabla_\theta\left[\boldsymbol{x}_k + \left(\boldsymbol{v}_\theta(\boldsymbol{x}_k, k) + \frac{\sigma_k^2}{2k}(\boldsymbol{x}_k + (1-k)\boldsymbol{v}_\theta(\boldsymbol{x}_k, k))\right)\Delta k\right] \tag{17}$$

$$= \nabla_\theta\left[\left(\boldsymbol{v}_\theta(\boldsymbol{x}_k, k) + \frac{\sigma_k^2(1-k)}{2k}\boldsymbol{v}_\theta(\boldsymbol{x}_k, k)\right)\Delta k\right] \tag{18}$$

$$= \left(1 + \frac{\sigma_k^2(1-k)}{2k}\right)\Delta k \cdot \nabla_\theta \boldsymbol{v}_\theta(\boldsymbol{x}_k, k) \tag{19}$$

Substituting back:

$$\nabla_\theta \mathcal{J}(\theta) = \sum_{k=0}^{T-1} \mathbb{E}_{\boldsymbol{x}_T \sim \mathcal{N}(0,\boldsymbol{I}), \boldsymbol{\epsilon} \sim \mathcal{N}(0,\boldsymbol{I})}\left[\frac{\boldsymbol{\epsilon}}{\sigma_k\sqrt{\Delta k}} \cdot \left(1 + \frac{\sigma_k^2(1-k)}{2k}\right)\Delta k \cdot \nabla_\theta \boldsymbol{v}_\theta(\boldsymbol{x}_k, k)\hat{A}_k\right] \tag{20}$$

$$= \sum_{k=0}^{T-1} \mathbb{E}_{\boldsymbol{x}_T \sim \mathcal{N}(0,\boldsymbol{I}), \boldsymbol{\epsilon} \sim \mathcal{N}(0,\boldsymbol{I})}\left[\left(\frac{\sqrt{\Delta k}}{\sigma_k} + \frac{\sigma_k\sqrt{\Delta k}(1-k)}{2k}\right) \cdot \boldsymbol{\epsilon} \cdot \nabla_\theta \boldsymbol{v}_\theta(\boldsymbol{x}_k, k)\hat{A}_k\right] \tag{21}$$

With $\sigma_k = a\sqrt{\frac{k}{1-k}}$, we get:

$$\frac{\sqrt{\Delta k}}{\sigma_k} = \frac{\sqrt{\Delta k}}{a\sqrt{\frac{k}{1-k}}} = \frac{1}{a}\sqrt{\frac{\Delta k(1-k)}{k}} \tag{22}$$

$$\frac{\sigma_k\sqrt{\Delta k}(1-k)}{2k} = \frac{a\sqrt{\frac{k}{1-k}}\sqrt{\Delta k}(1-k)}{2k} = \frac{a}{2}\sqrt{\frac{\Delta k(1-k)}{k}} \tag{23}$$

Therefore:

$$\nabla_\theta \mathcal{J}(\theta) = \sum_{k=0}^{T-1} \mathbb{E}_{\boldsymbol{x}_T \sim \mathcal{N}(0,\boldsymbol{I}), \boldsymbol{\epsilon} \sim \mathcal{N}(0,\boldsymbol{I})} \left[ \left(\frac{1}{a} + \frac{a}{2}\right) \underbrace{\sqrt{\frac{\Delta k(1-k)}{k}}}_{\text{Scale Term}} \cdot \boldsymbol{\epsilon} \cdot \nabla_\theta \boldsymbol{v}_\theta(\boldsymbol{x}_k, k)\hat{A}_k \right] \tag{24}$$

This reveals that the natural gradient coefficient is proportional to $\sqrt{\frac{1-k}{k}}\sqrt{\Delta k}$, which captures the intrinsic exploration potential at timestep $k$. After reweighting, we have the following derivation (don't consider $a$ and norm coefficient):

$$\nabla_\theta \mathcal{J}(\theta) = \sum_{k=0}^{T-1} \mathbb{E}_{\boldsymbol{x}_T \sim \mathcal{N}(0,\boldsymbol{I}), \boldsymbol{\epsilon} \sim \mathcal{N}(0,\boldsymbol{I})} \left[ \left(\frac{1}{a} + \frac{a}{2}\right) \underbrace{\Delta k}_{\text{Scale Term}} \cdot \boldsymbol{\epsilon} \cdot \nabla_\theta \boldsymbol{v}_\theta(\boldsymbol{x}_k, k)\hat{A}_k \right] \tag{25}$$

Consider $\mathbb{E}_{\boldsymbol{\epsilon} \sim \mathcal{N}(0,\boldsymbol{I})}[\boldsymbol{\epsilon}\hat{A}_k]$, suppose the final reward is a function of the small noise vector $\sigma_k\sqrt{\Delta k}\boldsymbol{\epsilon}$ applied at a certain step. When $\sigma_k\sqrt{\Delta k}\boldsymbol{\epsilon}$ is small (and drawn from a zero-mean Gaussian), we can approximate the reward using a first-order Taylor expansion:

$$R_k(\sigma_k\sqrt{\Delta k}\boldsymbol{\epsilon}) \approx R_k(0) + \sigma_k\sqrt{\Delta k}\boldsymbol{\epsilon}^T \nabla_{\sigma_k\sqrt{\Delta k}\boldsymbol{\epsilon}} R_k|_{\sigma_k\sqrt{\Delta k}\boldsymbol{\epsilon}=0} \tag{26}$$

Since $\hat{A}_k$ is normalized version of $R_k$, the mean and std are as follows (let $\nabla_{\sigma_k\sqrt{\Delta k}\boldsymbol{\epsilon}} R_k|_{\sigma_k\sqrt{\Delta k}\boldsymbol{\epsilon}=0} = g_k$):

$$
\begin{aligned}
\text{mean} &= \mathbb{E}_{\sigma_k\sqrt{\Delta k}\boldsymbol{\epsilon}}[R_k(0) + \sigma_k\sqrt{\Delta k}\boldsymbol{\epsilon}^T g_k] = R_k(0) \\
\text{std} &= \sqrt{\mathbb{E}_{\sigma_k\sqrt{\Delta k}\boldsymbol{\epsilon}}[(R_k(\sigma_k\sqrt{\Delta k}\boldsymbol{\epsilon}) - \text{mean})^2]} \\
&= \sqrt{\mathbb{E}_{\sigma_k\sqrt{\Delta k}\boldsymbol{\epsilon}}[(\sigma_k\sqrt{\Delta k}\boldsymbol{\epsilon}^T g_k)^2]} \\
&= \sigma_k\sqrt{\Delta k}||g_k||
\end{aligned}
\tag{27}
$$

Therefore:

$$
\begin{aligned}
\hat{A}_k &= \frac{R_k - \text{mean}}{\text{std}} = \frac{\sigma_k\sqrt{\Delta k}\boldsymbol{\epsilon}^T g_k}{\sigma_k\sqrt{\Delta k}||g_k||} \\
\mathbb{E}_{\boldsymbol{\epsilon}}[\boldsymbol{\epsilon}\hat{A}_k] &= \mathbb{E}_{\boldsymbol{\epsilon}}\left[\frac{\boldsymbol{\epsilon}\boldsymbol{\epsilon}^T g_k}{||g_k||}\right] = \frac{g_k}{||g_k||}
\end{aligned}
\tag{28}
$$

Equation 28 indicates that the norm of $\mathbb{E}_{\boldsymbol{\epsilon}}[\boldsymbol{\epsilon}\hat{A}_k]$ is invariant among the timesteps.

## A.2 EXPERIMENTAL SETTING DETAILS

**Dataset.** We evaluate Compositional Image Generation on Geneval and Human Preference Alignment on PickScore. Meanwhile, we also test our method on HPDv2 (Wu et al., 2023).

**Reward Model.** In TempFlow-GRPO, we introduce four reward models, including clip-based PickScore, HPSv2 (Wu et al., 2023), VLM-based HPSv3 (Ma et al., 2025b) and framework Geneval.

**Human Preference Alignment.** Following Flow-GRPO, we evaluate our method on PickScore prompts using the PickScore reward function. PickScore is a CLIP-based scoring function that demonstrates superhuman performance in predicting human preferences for generated images. To ensure consistency with Flow-GRPO, we set the KL divergence regularization coefficient $\beta = 0.001$.

**Compositional Image Generation.** We adopt the GenEval framework following Flow-GRPO's experimental protocol. The training prompts are sourced from the Flow-GRPO dataset. GenEval provides an object-focused evaluation framework that assesses compositional image properties, including object co-occurrence, spatial positioning, object count, and color attributes. For fair comparison, we maintain the KL divergence regularization coefficient at $\beta = 0.004$.

**HPSv3 Evaluation on FLUX.1-dev.** For experiments on the FLUX.1-dev model, we configure the noise level at 0.9 for both Flow-GRPO and our TempFlow-GRPO method. The KL divergence weight is set to 0.004. During training, we employ 10 sampling steps, while evaluation uses 50 steps to ensure high-quality generation. The batch configuration consists of a group size of 24 (arranged as $4 \times 6$) with 48 groups in total.

**Comparison with DanceGRPO.** DanceGRPO and Flow-GRPO represent concurrent developments in this research area. The SDE sampling methodology employed in DanceGRPO corresponds to Equation 12 in the Flow-GRPO formulation. However, the theoretical derivation in DanceGRPO terminates at its Equation 7 without developing an analogous final formulation. To comprehensively validate our method's effectiveness, we conduct a direct comparison with DanceGRPO. For equitable comparison, we train TempFlow-GRPO for 300 iterations on the HPDv2 dataset using the HPSv2 reward function.

### A.3 Experiments on FLUX.1-dev

We further evaluate the performance of our method against Flow-GRPO on FLUX.1-dev, utilizing the HPDv2 dataset with HPSv3 as the reward model. As illustrated in Figure 10, our method demonstrates substantial improvements over the baseline. After 300 training steps, our approach achieves a performance gain of approximately 0.3 compared to Flow-GRPO. More significantly, our method exhibits superior training efficiency: it reaches the performance level that Flow-GRPO achieves at 300 steps in merely 80 steps, representing a $3.75\times$ speedup in convergence. Additionally, the results indicate that our method maintains lower and more stable KL divergence values throughout the optimization process, suggesting improved training dynamics and better preservation of the original model's distribution compared to Flow-GRPO.

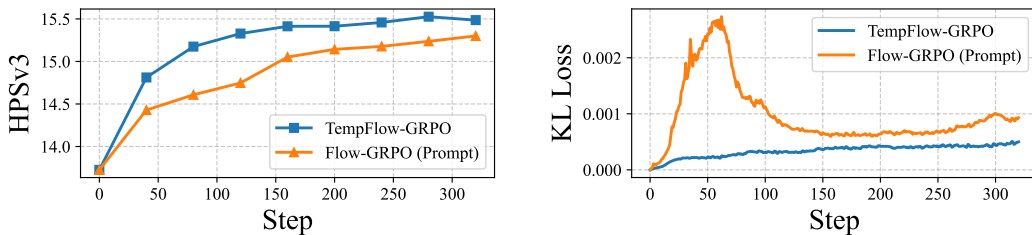

Figure 10: The performance and KL loss (smoothed) of HPSv3 on HPDv2 dataset with FLUX.1-dev as base model.

### A.4 Comparsion with DanceGRPO

As illustrated in Figure 11 (Left), we present the evaluation curves of our method on the HPDv2 dataset. The results demonstrate clear superiority of our approach over the baseline. While Dance-GRPO achieves a final performance score of 37.2 after 300 iterations, our method attains a higher score of 38.5 at 280 iterations, representing a 1.3% improvement over the DanceGRPO. More significantly, our approach exhibits substantially enhanced training efficiency: it matches DanceGRPO's final performance (achieved at 300 iterations) in merely 150 iterations, yielding a $2\times$ speedup in convergence. Notably, this superior performance is achieved without any specialized hyperparame-

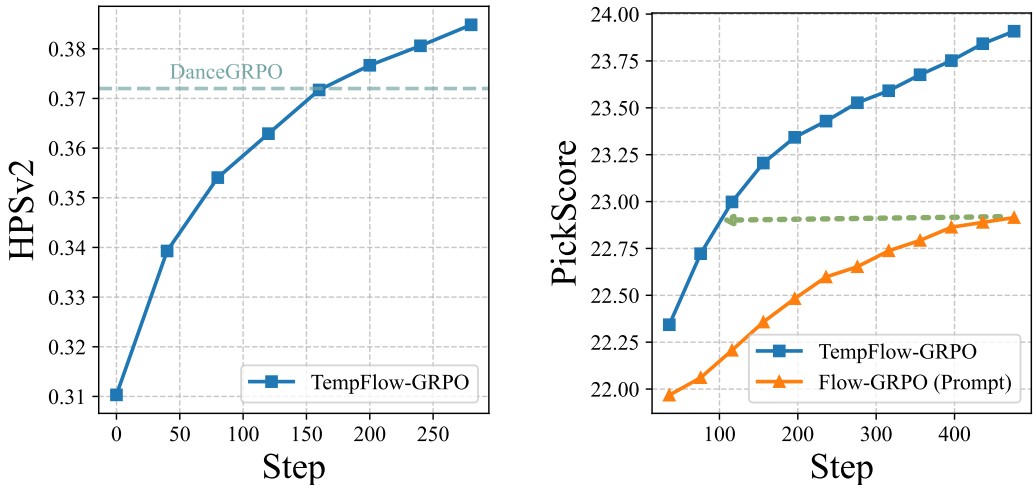

Figure 11: (Left) Performance of TempFlow-GRPO on HPDv2 with HPSv2 as reward compared DanceGRPO. (Right) Comparsion on PickScore benchmark (PickScore reward, SD3.5-M 1024).

ter tuning, as we directly adopt the configuration settings from Flow-GRPO, further validating the robustness and generalizability of our method.

## A.5 EXPERIMENTS ON SD3.5-M WITH 1024×1024 RESOLUTION

We investigate the effectiveness of our approach at higher image resolutions, employing PickScore as the reward model for evaluation. As illustrated in Figure 11 (Right), TempFlow-GRPO achieves a 1.0% improvement in PickScore after 450 training steps compared to the Flow-GRPO. More notably, our method demonstrates exceptional training efficiency, requiring only approximately 100 steps to match the performance that Flow-GRPO (Prompt) achieves after 450 steps—a 4.5× speedup. This result further validates the efficiency and effectiveness of our proposed method across varying image resolutions.

## A.6 TRAINING HOURS

A discussion of the computational cost of our method is warranted. Due to the $K$-branch exploration at each timestep, our sampling process incurs higher computational overhead compared to Flow-GRPO. Specifically, for $K = 10$, the average number of branches is approximately $(9 + \ldots + 1)/10 = 4.5$ times that of Flow-GRPO. However, the training time per iteration remains identical to Flow-GRPO. Despite this increased per-sample computational cost, our method demonstrates substantially superior overall training efficiency. As illustrated in Figure 3 (bottom left, main paper) and Figure 12, where we plot performance against wall-clock training time, our method exhibits faster convergence across all evaluated benchmarks. This efficiency gain becomes particularly pronounced at higher resolutions, where our approach achieves Flow-GRPO's final performance using only 33% to 50% of the total training time.

Furthermore, our approach not only converges faster but also achieves superior final performance. The PickScore experiments presented in the main paper reveal that even after extended training of 1200 steps, our method continues to outperform Flow-GRPO, providing compelling evidence that our approach possesses a higher performance ceiling.

## A.7 ADDITIONAL DETAILS ON GROUP STRATEGY

In Section 4.3 of the main paper, we analyze the impact of the group strategy on performance. Figure 6 and Figure 13 demonstrate that TempFlow-GRPO achieves substantial improvements even when employing an identical group strategy to the baseline, indicating the inherent advantages of our proposed core modules. To systematically disentangle the contributions of the group strategy

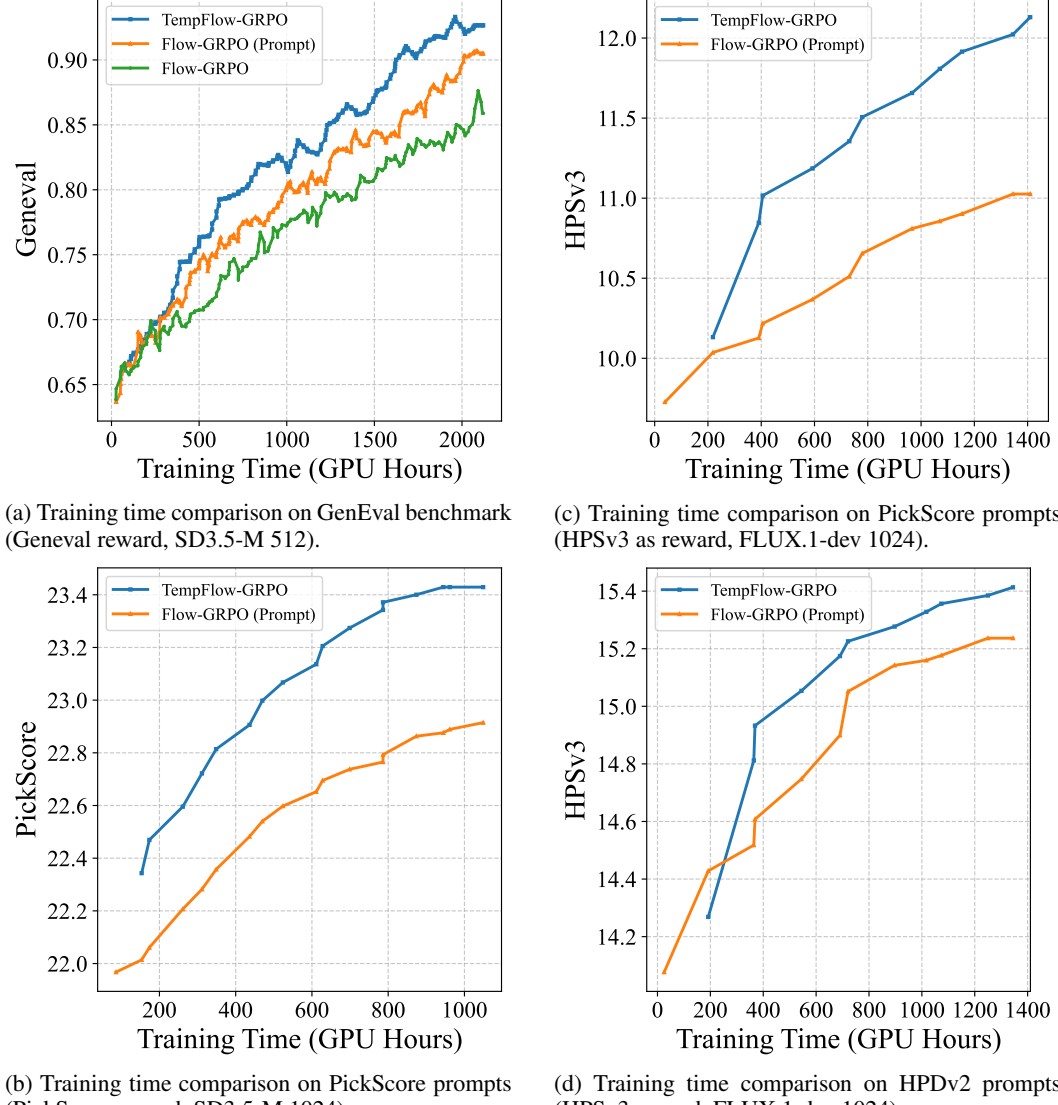

(a) Training time comparison on GenEval benchmark (Geneval reward, SD3.5-M 512).

(c) Training time comparison on PickScore prompts (HPSv3 as reward, FLUX.1-dev 1024).

(b) Training time comparison on PickScore prompts (PickScore reward, SD3.5-M 1024).

(d) Training time comparison on HPDv2 prompts (HPSv3 reward, FLUX.1-dev 1024).

Figure 12: Comprehensive comparison of training time across multiple benchmarks and resolutions. TempFlow-GRPO consistently achieves superior performance with reduced training time compared to Flow-GRPO baseline.

from our two fundamental components—trajectory branching and noise reweighting—we conduct controlled ablation studies. In these experiments, we maintain a constant group strategy configuration to ensure fair comparison. The results unequivocally demonstrate that our core modules independently yield significant performance gains. These components not only enable the model to match Flow-GRPO's multi-step performance in substantially fewer iterations but also elevate the final performance ceiling, validating the effectiveness of our theoretical framework.

## A.8 KL DIVERGENCE ANALYSIS

To further substantiate the training stability of our method, we provide a comprehensive analysis of the KL divergence dynamics during optimization. The results, presented in Figure 14, clearly demonstrate that our method consistently maintains significantly lower KL divergence values throughout the training process compared to Flow-GRPO. This reduced divergence indicates better preservation of the original model distribution while still achieving effective preference alignment.

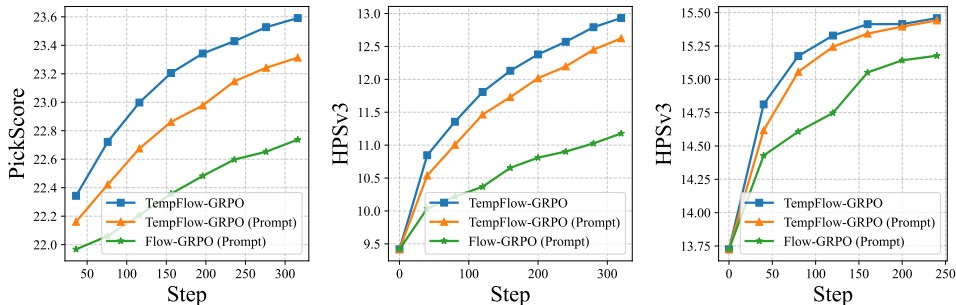

(a) Group strategy analysis on PickScore prompts (PickScore as reward, SD3.5-M 1024).

(b) Group strategy analysis on PickScore prompts (HPSv3 as reward, FLUX.1-dev 1024).

(c) Group strategy analysis on HPDv2 prompts (HPSv3 as reward, FLUX.1-dev 1024).

Figure 13: Comprehensive analysis of group strategy impact across different models and benchmarks, demonstrating consistent performance improvements with our proposed method and seed group approach.

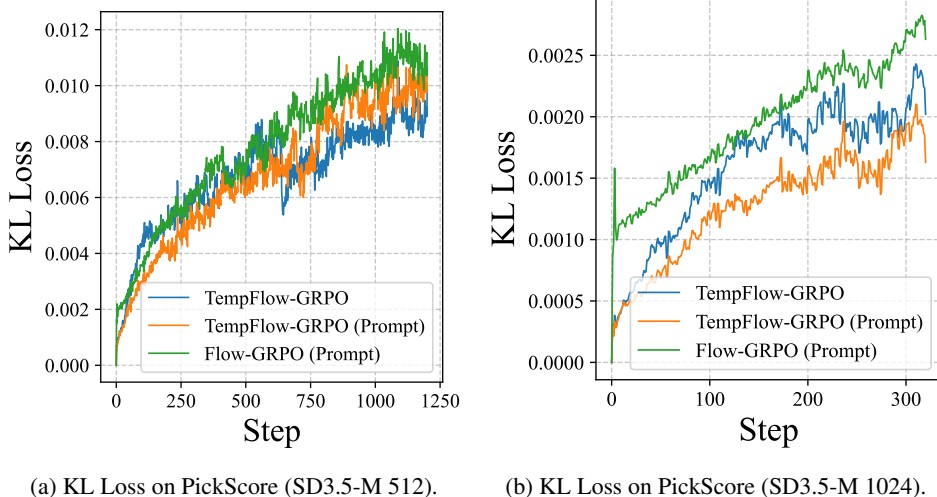

(a) KL Loss on PickScore (SD3.5-M 512).

(b) KL Loss on PickScore (SD3.5-M 1024).

Figure 14: Comprehensive analysis of KL divergence dynamics during training across different resolutions. TempFlow-GRPO consistently maintains lower KL divergence values compared to Flow-GRPO, indicating better preservation of the original model distribution while achieving superior preference alignment.

Furthermore, we analyze the impact of different grouping strategies on KL divergence. While adopting the "seed group" strategy results in a marginal increase in KL divergence compared to our default configuration, the values remain substantially below those of the Flow-GRPO baseline across all training iterations. This observation provides strong evidence that our core modules—trajectory branching and noise reweighting—inherently enhance training stability and robustly maintain distributional proximity, regardless of the specific grouping strategy employed. The consistent stability across different configurations underscores the fundamental advantages of our theoretical framework in balancing exploration and exploitation during the optimization process.

## A.9 MULTI-REWARD OPTIMIZATION

We evaluate the performance of our method in multi-reward optimization scenarios. For this experiment, we employ FLUX.1-dev as the base model and train at a high resolution of 1024×1024. We simultaneously optimize for two reward functions—HPSv3 and PickScore—with a weight ratio of 1:0.26. After 120 training steps, the results presented in Table 2 demonstrate our method's ro-

Table 2: Performance comparison for multi-reward optimization on PickScore prompts, demonstrating simultaneous improvement across multiple objectives.

|  | HPSv3 ↑ | PickScore ↑ |
|---|---|---|
| FLUX.1-dev | 9.42 | 22.34 |
| Flow-GRPO (Prompt) | 10.41 (+0.99) | 22.44 (+0.1) |
| TempFlow-GRPO | 12.23 (+1.81) | 22.52 (+0.18) |

bust capability for multi-objective optimization. Compared to Flow-GRPO, our approach achieves a substantial improvement of 0.82 on the HPSv3 metric while simultaneously maintaining consistent performance gains on PickScore. These results validate that our TempFlow-GRPO effectively balances multiple objectives without sacrificing performance on individual metrics, highlighting the versatility of our approach in complex preference alignment scenarios where multiple criteria must be optimized concurrently.

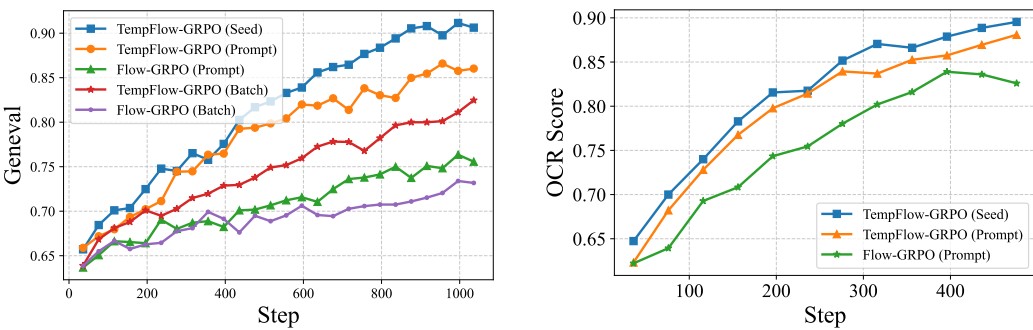

Figure 15: (Left) Performance of TempFlow-GRPO and Flow-GRPO on different group strategies (Geneval reward, SD3.5-M 512). (Right) Comparsion on Visual Text Rendering benchmark (OCR reward, SD3.5-M 512).

## A.10 GROUP STRATEGY ON GENEVAL

To further validate the robustness of TempFlow-GRPO across different group strategies, we conduct additional experiments on the GenEval benchmark. As illustrated in the left panel of Figure 15, the results demonstrate significant improvements with TempFlow-GRPO. Specifically, under the batch grouping strategy, TempFlow-GRPO achieves a performance gain of approximately 7% within 1,000 steps. This margin expands to around 10% when employing the prompt group strategy. Most notably, our proposed seed group strategy yields superior performance, attaining a GenEval score of 91% at 1,000 steps. These findings conclusively demonstrate that TempFlow-GRPO is a robust and highly effective method, regardless of the chosen group strategy.

## A.11 VISUAL TEXT RENDERING

To further demonstrate the superiority of TempFlow-GRPO, we conduct a comparative evaluation on the Visual Text Rendering task.

**Experimental Setup.** Following the configuration in Flow-GRPO, we use SD3.5-M as the base model with a resolution of 512×512. The noise level is set to 0.7, and the KL loss weight is set to 0.004. Additionally, we employ PaddleOCR as the reward model. We evaluate both the prompt grouping and our proposed seed grouping strategies.

As illustrated in the right panel of Figure 15, TempFlow-GRPO exhibits significant performance gains on this task. Specifically, TempFlow-GRPO (using the prompt grouping strategy) outperforms the Flow-GRPO baseline, achieving an improvement of approximately 5.5% at around 500 steps. Furthermore, our seed grouping variant attains the highest performance, reaching a score of ap-

proximately 89.5%, which surpasses the Flow-GRPO baseline by 7%. These results underscore the effectiveness and versatility of the proposed TempFlow-GRPO framework.

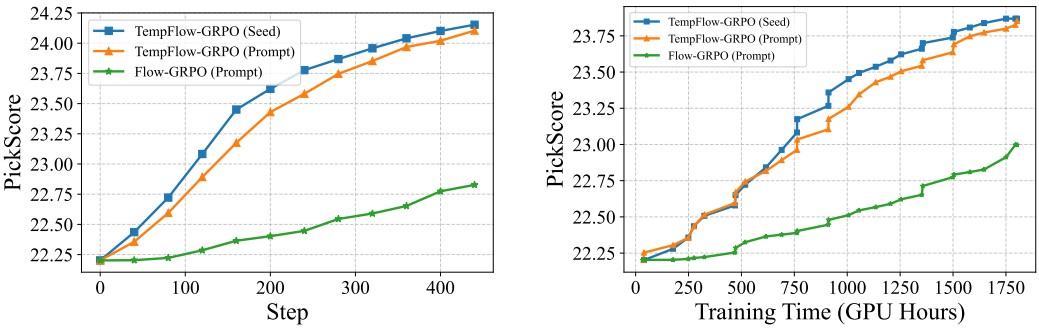

Figure 16: (Left) Performance comparison on the PickScore benchmark across training steps (Qwen-Image 512). (Right) Performance comparison on the PickScore benchmark across GPU hours (Qwen-Image 512).

## A.12 TEMPFLOW-GRPO ON QWEN-IMAGE

We validate TempFlow-GRPO on Qwen-Image Wu et al. (2025a). The experimental setup involves LoRA fine-tuning with a KL weight of 0 and a noise level of 1.2, utilizing PickScore as the reward model (at a resolution of 512).

The results, illustrated in Figure 16, highlight significant improvements. Notably, even without additional hyperparameter tuning, our method (configured with a 4×6 group size and simple reweighting) substantially outperforms Flow-GRPO (prompt). It achieves a 1.3-point performance gain over the baseline. Critically, TempFlow-GRPO matches the performance level that Flow-GRPO achieves at 400 steps in only 100 steps. Regarding computational cost, our method requires only 43% of the total training time. This confirms the high efficiency and ease of integration of our proposed approach.

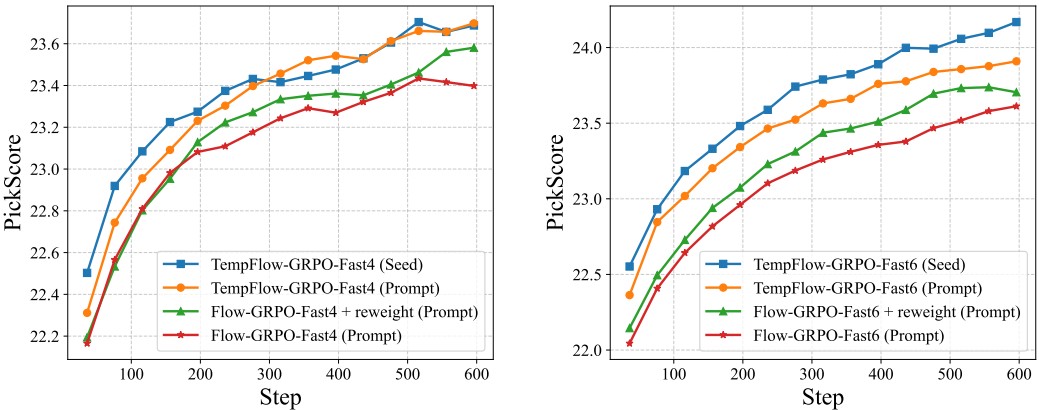

Figure 17: (Left) Performance of TempFlow-GRPO-Fast and Flow-GRPO-Fast in 4 steps (PickScore, SD3.5-M 512). (Right) Performance of TempFlow-GRPO-Fast and Flow-GRPO-Fast in 6 steps (PickScore, SD3.5-M 512).

## A.13 HYBRID VARIANTS OF TEMPFLOW-GRPO

In this section, we introduce a variant of Flow-GRPO, **Flow-GRPO-Fast**, designed to accelerate the training process. This is achieved by strategically limiting the SDE formulation to the initial timesteps of the diffusion process. Figure 17 illustrates the performance of two configurations:

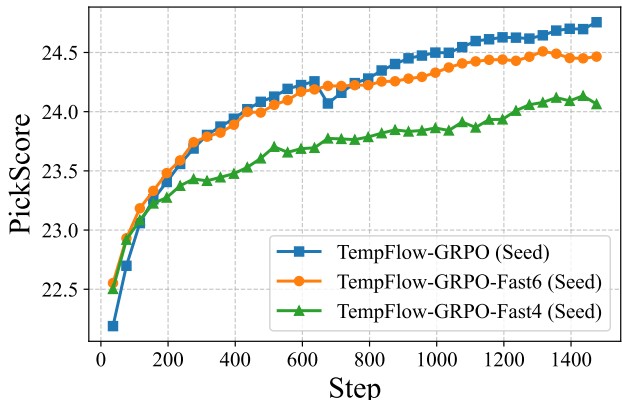

Figure 18: Performance of TempFlow-GRPO-Fast in different steps.

Flow-GRPO-Fast4 and Flow-GRPO-Fast6, which correspond to training with the first 4 and 6 steps, respectively.

**Reweighting on Flow-GRPO-Fast.** To validate the effectiveness of our reweighting strategy, we conduct an ablation study. In this experiment, we omit the trajectory branching and apply the reweighting strategy directly to Flow-GRPO-Fast. The results are presented in Figure 17, where the left panel shows the 4-step setting and the right panel displays the 6-step setting. As observed, applying the reweighting strategy alone yields considerable performance improvements, particularly in these few-step settings. Notably, this performance gain is achieved at **zero additional computational cost**, as the reweighting mechanism introduces no overhead to either the training or sampling time.

**TempFlow-GRPO-Fast.** Next, we introduce a variant named **TempFlow-GRPO-Fast** to evaluate the effectiveness of our framework in few-step settings. This variant demonstrates the robustness of TempFlow-GRPO across varying numbers of steps. As depicted in Figure 17, the results confirm consistent improvements. In the 4-step setting, TempFlow-GRPO-Fast (Prompt) and TempFlow-GRPO-Fast (Seed) both achieve a performance gain of 0.3. These improvements become more pronounced in the 6-step setting, where the Prompt and Seed strategies yield gains of 0.3 and 0.5, respectively. These findings validate that TempFlow-GRPO maintains its performance superiority even in computationally constrained, few-step settings. Note that reducing the number of steps negatively impacts overall performance, as illustrated in Figure 18. This is because each step in the diffusion process contributes to the exploration of the solution space.

## A.14 THE DIVERSITY OF TEMPFLOW-GRPO

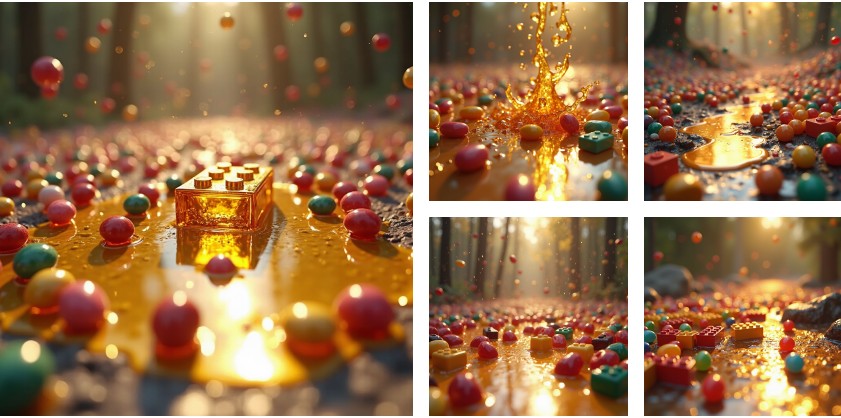

Figure 19: The diversity of TempFlow-GRPO on FLUX.1-dev (Prompt: "The ground is a mixture of melting liquid gold and scattered LEGO bricks. It is raining colorful jelly beans from the sky. Macro photography, shallow depth of field, hyper-realistic light refraction, tactile textures, unreal engine 5 render, volumetric lighting").

To demonstrate the diversity of our approach, we present visualizations in Figure 19. The results indicate that TempFlow-GRPO maintains high diversity while achieving superior prompt alignment and image quality. To assess potential reward hacking, we evaluate the PickScore metric on models optimized for the GenEval benchmark. After approximately 3,800 steps, the scores for both our method and the baseline decrease by approximately 0.234, which indicates that our approach does not introduce additional reward hacking issues. Furthermore, TempFlow-GRPO exhibits lower and more stable KL divergence. We acknowledge that reward hacking remains an inherent challenge in RL-based strategies, and we plan to address this issue from both algorithmic and reward-modeling perspectives in future work.

## A.15 ALGORITHM OF TEMPFLOW-GRPO

Algorithm 1 details the operational workflow of Tempflow-GRPO for a single input prompt, specifically configured with a $N \times M$ group size.

---

**Algorithm 1** TempFlow-GRPO (Take a prompt $c$ as case)

---

**Require:** Prompt $c$, group size $G(N \times M)$, total timesteps $T$, reward models $R$.
**Ensure:** Optimized policy parameters $\theta$
 1: Initialize policy parameters $\theta$, reference policy $\pi_{\text{ref}}$
 2: **repeat**
 3:    // Do this on N GPUs
 4:    // Sample
 5:    Sample initial noise $\boldsymbol{x}_T \sim \mathcal{N}(0, I)$ // Ensure the same initial noise
 6:    **for** $t = T$ **to** $0$ **do**
 7:       $\boldsymbol{x}_{t-1} = \boldsymbol{x}_t - \boldsymbol{v}_t dt$ // Equation 4
 8:       // Traj Branch, M is the branch / batch size
 9:       **for** $i = 1$ **to** $M$ **do**
10:          $\hat{\boldsymbol{x}}_{t-1}^i = \boldsymbol{x}_t + [\boldsymbol{v}_\theta(\boldsymbol{x}_t, t) + \frac{\sigma_t^2}{2t}(\boldsymbol{x}_t + (1-t)\boldsymbol{v}_\theta(\boldsymbol{x}_t, t))]\Delta t + \sigma_t\sqrt{\Delta t}\boldsymbol{\epsilon}^i$ // Equation 5
11:          **for** $k = T - 1$ **to** $0$ **do**
12:             $\hat{\boldsymbol{x}}_{k-1}^i = \hat{\boldsymbol{x}}_k^i - \boldsymbol{v}_k dk$ // Equation 4
13:          **end for**
14:       **end for**
15:       Compute rewards $\{R(\hat{\boldsymbol{x}}_0^i, \boldsymbol{c})_t\}_{i=1}^M$
16:    **end for**
17:    // Compute Advantages
18:    **for** $t = T$ **to** $0$ **do**
19:       Compute mean, std of $\{R(\hat{\boldsymbol{x}}_0^i, \boldsymbol{c})_t\}_{i=1}^M$ // The group is initialized with same noise
20:       Compute $\{A(\hat{\boldsymbol{x}}_0^i, \boldsymbol{c})_t\}_{i=1}^M$ with Equation 3
21:    **end for**
22:    // Training
23:    **for** $i = 1$ **to** $M$ **do**
24:       **for** $t = T$ **to** $0$ **do**
25:          // Policy Reweighting
26:          $\mathcal{L}_t^i = \mathcal{J}_{\text{policy}_t}^i - D_{\text{KL}_t}^i$ with Equation 6 and Equation 7
27:       **end for**
28:    **end for**
29:    $\mathcal{L}_{\text{total}} = \frac{1}{M \cdot T} \sum_{i=1}^M \sum_{t=1}^T \mathcal{L}_t^i$
30:    $\theta \leftarrow \theta - \eta \nabla_\theta \mathcal{L}_{\text{total}}$
31: **until** convergence

---

## A.16 GENERALIZED FORMULATION OF REWEIGHTING STRATEGY

To provide a more general understanding of our approach, we derive the generalized reweighting coefficient $\beta$ for arbitrary $\sigma_k$ schedules. Based on Equation 21 and Equation 25, we obtain:

$$\beta \left( \frac{\sqrt{\Delta k}}{\sigma_k} + \frac{\sigma_k \sqrt{\Delta k}(1-k)}{2k} \right) = \left( \frac{1}{a} + \frac{a}{2} \right) \Delta k \tag{29}$$

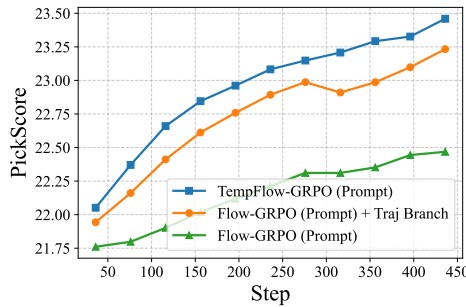

Figure 20: Performance of TempFlow-GRPO under $\sigma_k = ak$ (PickScore, SD3.5-M, 512).

Solving for $\beta$ yields:

$$\beta = \left(\frac{1}{a} + \frac{a}{2}\right) \frac{\sqrt{\Delta k}}{\frac{1}{\sigma_k} + \frac{\sigma_k(1-k)}{2k}} \tag{30}$$

$$\beta = \left(\frac{1}{a} + \frac{a}{2}\right) \frac{2k\sigma_k\sqrt{\Delta k}}{2k + \sigma_k^2(1-k)} \tag{31}$$

In the specific case of Flow-GRPO, where $\sigma_k = a\sqrt{\frac{k}{1-k}}$, this simplifies to:

$$\beta = \sqrt{\frac{k}{1-k}}\sqrt{\Delta k} \tag{32}$$

Note that Flow-GRPO determines its optimal $\sigma_k$ by specifically formulating it as the Signal-to-Noise Ratio (SNR). According to Equation 31, our method applies to any $\sigma_k$. Consequently, our weighting coefficient remains effective across varying formulations. We demonstrate this flexibility in Figure 20, where our method consistently outperforms the baseline even when $\sigma_k$ is set to $ak$.

### A.17 VISUALIZATION

We present comprehensive qualitative comparisons to demonstrate the superior visual quality achieved by our TempFlow-GRPO method. The visualization results consistently show that our approach generates images with enhanced fidelity, better adherence to complex prompts, and fewer visual artifacts compared to baseline methods.

**Prompts in Figure 1.** The prompts in Figure 1 are as follows:

1. 16-year-old teenager wearing a white bear-ear hat with a smirk on their face.
2. photo of well done salmon dinner, 8K, Global Illumination, Ray Tracing Reflections
3. A lemon with a McDonald's hat.
4. An image of an emo with dark brown hair in a messy pixie cut, large entirely-black eyes, wearing black clothing.
5. The image is a mixed media collage with broken glass and torn paper elements, featuring intricate oil details and a canvas texture, in a contemporary art style.
6. An epic deep space photograph. Gigantic, monolithic letters forming the word 'TempFlow-' hang silently in the void, their surfaces like ancient, cracked obsidian reflecting distant starfields. Far below, the letters 'GRPO' are formed by a vast, tranquil nebula glowing with soft, ethereal light, like a cosmic ocean. The sense of scale is immense and humbling. Perfect, case-sensitive lettering. Moody, atmospheric, photorealistic, cinematic wide-angle shot, 4K UHD.
7. Kiwi fruit, mint leaves, ice cubes, background yellow, splashing water, soft box, back light, creative food photography, Art by Alberto Seveso,
8. Claymation of Futurama characters.
9. A group of four friends commemorating a ski trip in the snow.
10. an empty bench next to a busy street.
11. a 12 year old girl and her pet raccoon

FLUX.1-dev          Flow-GRPO (Prompt)          TempFlow-GRPO

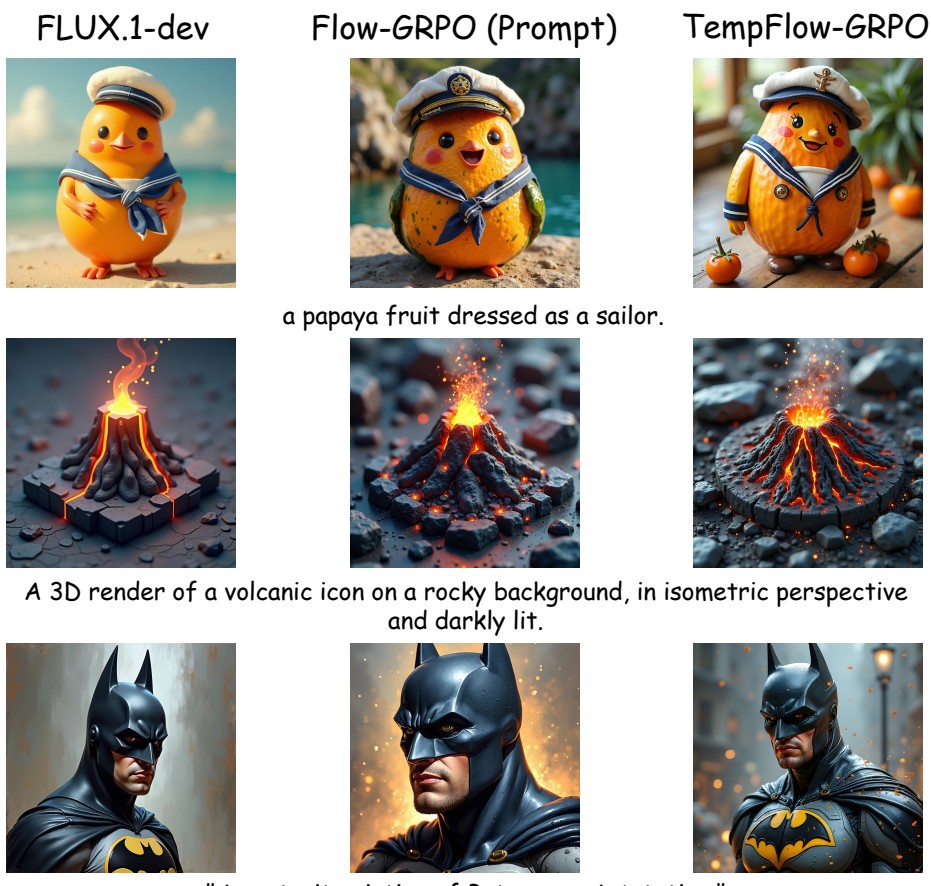

a papaya fruit dressed as a sailor.

A 3D render of a volcanic icon on a rocky background, in isometric perspective and darkly lit.

"A portrait painting of Batman on Artstation."

Figure 21: Qualitative comparison between FLUX.1-dev, Flow-GRPO (Prompt) and TempFlow-GRPO with HPSv3 rewards on HPDv2 prompts.

FLUX.1-dev          Flow-GRPO (Prompt)          TempFlow-GRPO

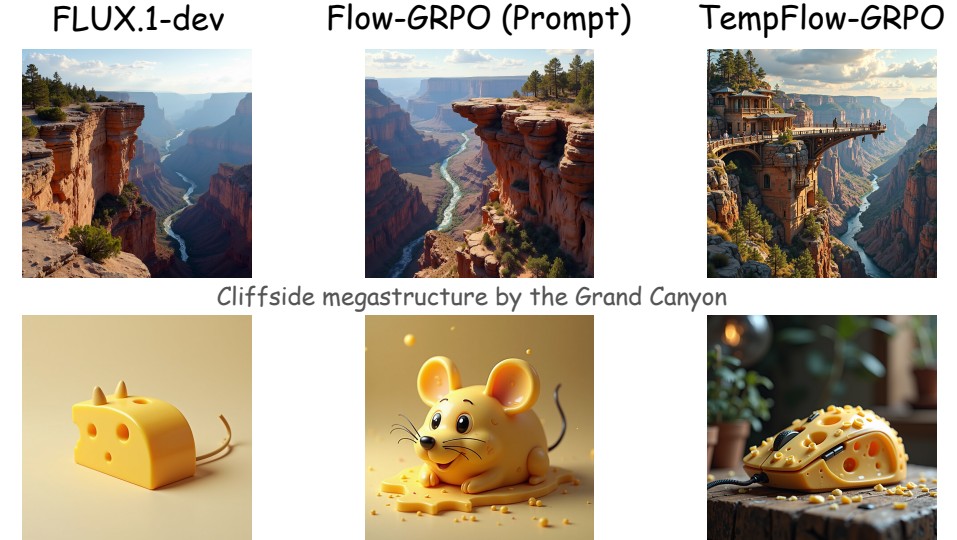

Cliffside megastructure by the Grand Canyon

render of pc mouse made out of cheese

Figure 22: Qualitative comparison between FLUX.1-dev, Flow-GRPO (Prompt) and TempFlow-GRPO with multi rewards on PickScore prompts.

FLUX.1-dev     Flow-GRPO (Prompt)     TempFlow-GRPO

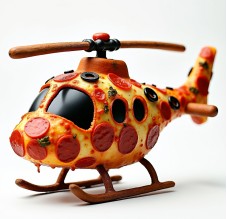 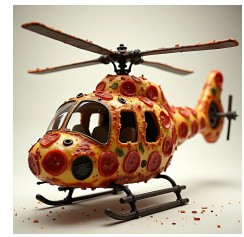 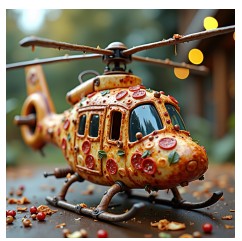

a whimsical and playful image of a pizza helicopter with toppings like pepperoni, mushrooms, peppers, and olives. The pizza should be transformed into the shape of a helicopter, complete with rotors, landing gear, and cockpit windows.

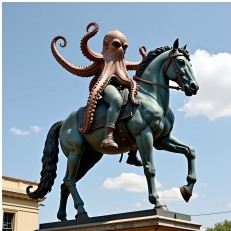 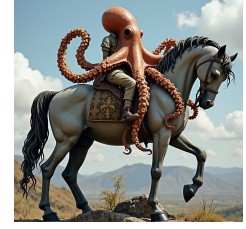 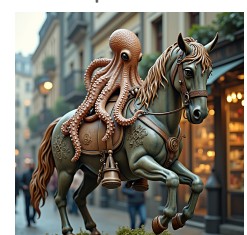

octopus on top of a horse statue

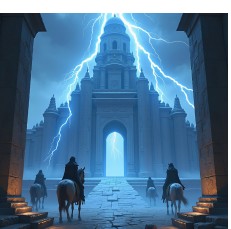 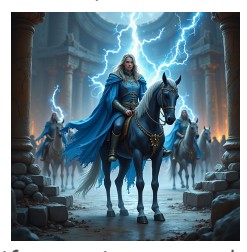 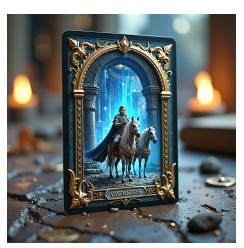

Lightning Greaves: This artifact equipment card gives the equipped creature haste and shroud, making it a popular choice for protecting and enhancing creatures in combat. Its low cost of 2 mana also makes it a versatile card in many decks...

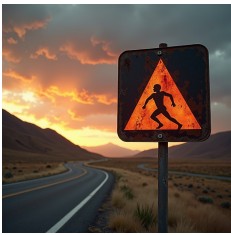 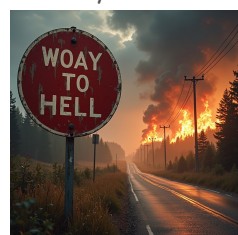 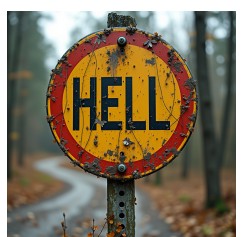

A road sign showing the way to hell

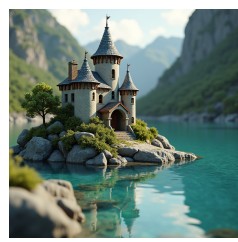 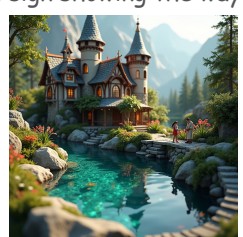 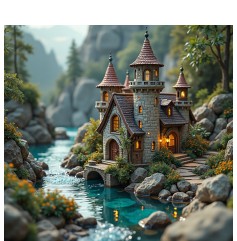

miniature fantasy castle with pool of water, sharp focus, photo taken with eos 5d, ultra realism, hyperrealism, professional photography, 8k uhd, ray tracing, ssao, film grain, long shot, wide shot

Figure 23: Qualitative comparison between FLUX.1-dev, Flow-GRPO (Prompt) and TempFlow-GRPO with multi rewards on PickScore prompts.

| FLUX.1-dev | Flow-GRPO (Prompt) | TempFlow-GRPO |
|:---:|:---:|:---:|
| 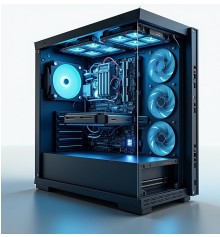 | 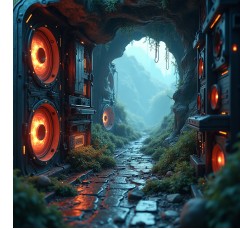 | 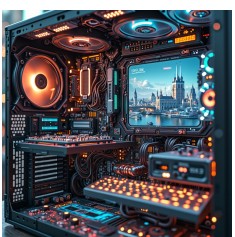 |

Inside a computer, high tech landscape, tron, intricately detailed, best quality

| 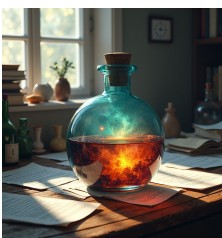 | 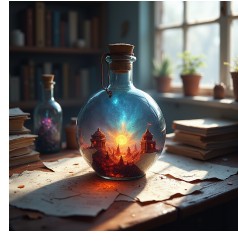 | 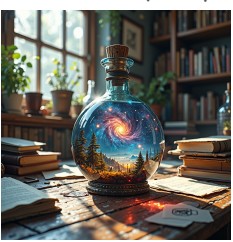 |
|:---:|:---:|:---:|

arcane style, one single colorful potion in a round bottle with a glowing galactic landscape inside of it on a messy brown table, papers and books, sunlight from a window, soft lighting, atmospheric, bottle is the focus. by makoto shinkai, stanley artgerm lau, wlop, rossdraws, james jean, andrei riabovitchev, marc simonetti, krenz cushart, sakimichan,

| 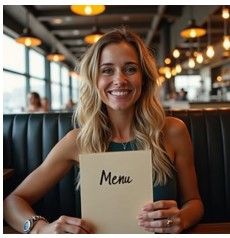 | 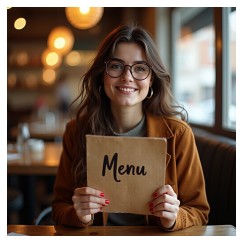 | 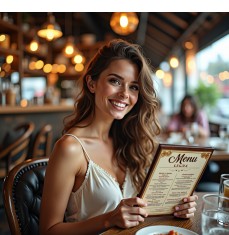 |
|:---:|:---:|:---:|

Photo of a woman sitting in a restaurant holding a menu that says "Menu"

| 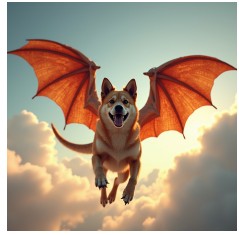 | 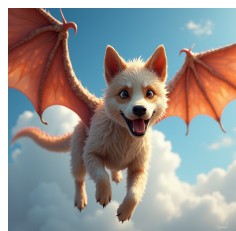 | 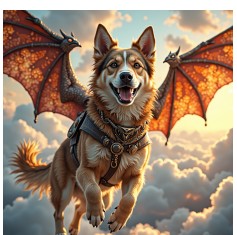 |
|:---:|:---:|:---:|

An image hyper realistic of a dog in the sky with dragon wings

| 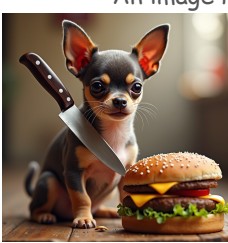 | 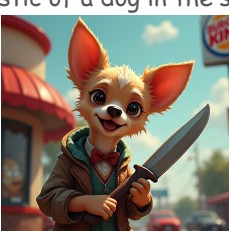 | 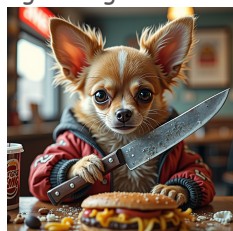 |
|:---:|:---:|:---:|

a chihuahua with a knife on a Burger King

Figure 24: Qualitative comparison between FLUX.1-dev, Flow-GRPO (Prompt) and TempFlow-GRPO with multi rewards on PickScore prompts.

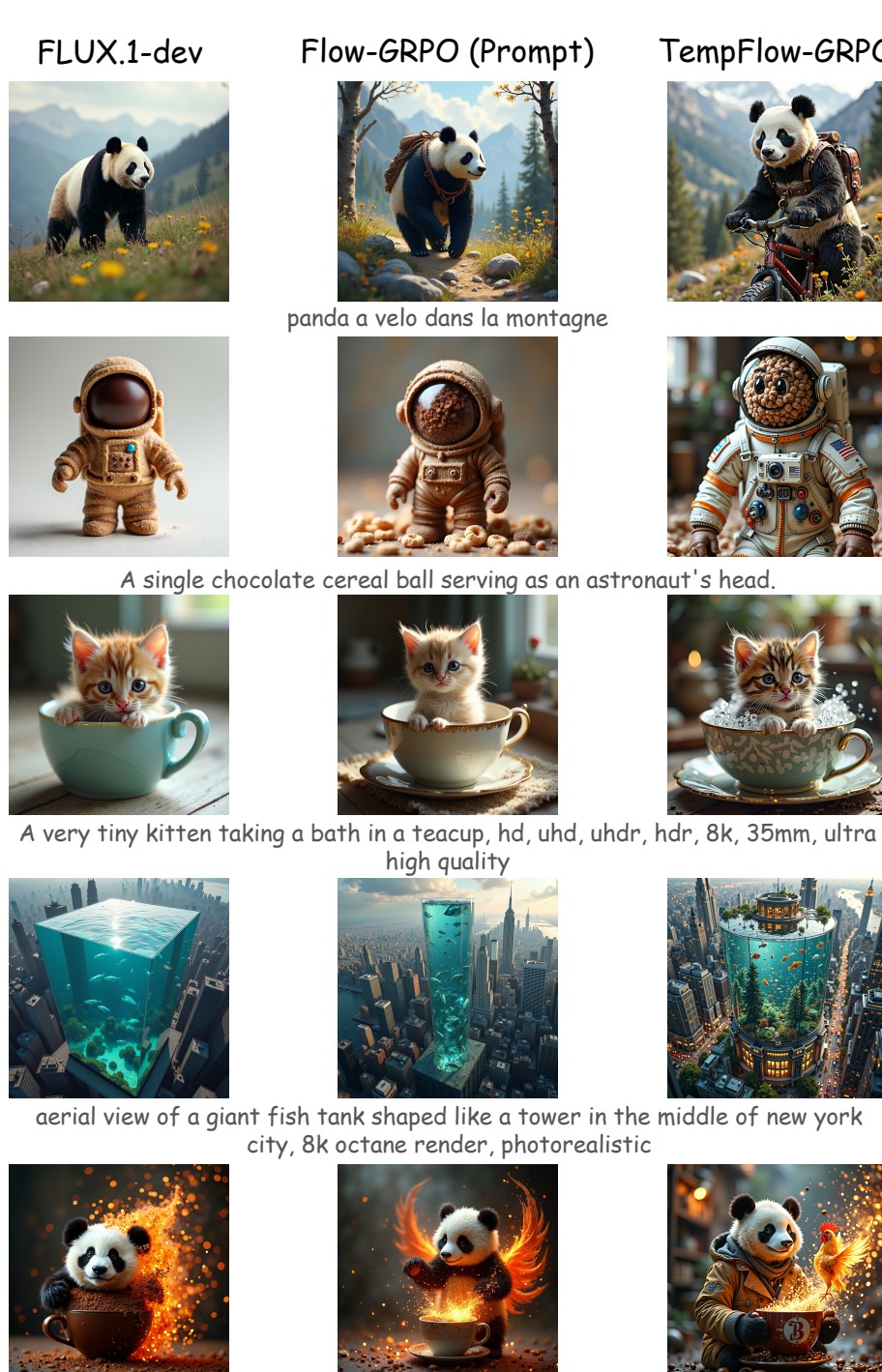

FLUX.1-dev    Flow-GRPO (Prompt)    TempFlow-GRPO

panda a velo dans la montagne

A single chocolate cereal ball serving as an astronaut's head.

A very tiny kitten taking a bath in a teacup, hd, uhd, uhdr, hdr, 8k, 35mm, ultra high quality

aerial view of a giant fish tank shaped like a tower in the middle of new york city, 8k octane render, photorealistic

A panda bear as a mad scientista cup of coffee splashing morph into a rising phoenix rooster with bramble and sparkle sparks spark particles, beautiful studio lighting, appetizing lighting

Figure 25: Qualitative comparison between FLUX.1-dev, Flow-GRPO (Prompt) and TempFlow-GRPO with multi rewards on PickScore prompts.

