# OpenReview forum: "TEMPFLOW-GRPO: WHEN TIMING MATTERS FOR GRPO IN FLOW MODELS"
_ICLR.cc/2026/Conference — ICLR 2026 Poster_

### Official Review · Reviewer_rGTd · 2025-10-25

**Soundness:** 4
**Presentation:** 4
**Contribution:** 4
**Rating:** 10
**Confidence:** 4

**Summary:**

The paper proposed TempFlow-GRPO, a framework that makes the optimization process temporally aware to address the key limitation of temporal uniformity in previous RLHF works. The paper introduces a mixture of ODE and SDE sampling, along with a noise-aware policy weighting scheme, to balance exploration and reward exploitation. Experiments demonstrate that TempFlow-GRPO achieves state-of-the-art performance, yielding higher rewards than standard GRPO approaches.

**Strengths:**

- The paper pinpoints temporal uniformity as the primary limitation of existing flow-based GRPO methods and proposes TempFlow-GRPO to solve it with precise credit assignment and noise-aware optimization. The authors demonstrate this non-uniformity well with empirical evidence from rewards, supporting the need for temporal information.
- The paper introduces the core mechanisms of trajectory branching and noise-aware reweighting to create temporally-structured policies that respect the dynamics of the generative process. The authors also provide a theoretical justification from the policy gradient perspective, further supporting the use of noise-aware reweighting.
- The proposed TempFlow-GRPO achieves state-of-the-art performance compared to the existing vanilla GRPO approach, demonstrating the effectiveness of the method. The authors also include comprehensive ablation studies to better understand the dynamics of this model.

**Weaknesses:**

- The computational cost, as thoroughly analyzed in Appendix A.6, will be higher than the vanilla GRPO models due to the branching process. Nonetheless, this is more like a trade-off between quality and time, given the superior quality metrics.

**Questions:**

- How is the performance affected by the number of branches (K) at each step, the specific timesteps chosen for branching, or the exact function used for noise-aware weighting? The ablation study (Fig. 8) shows that the 4x6 (seed x branch) configuration was chosen, but it's unclear how much tuning is required to find the optimal setup for a new model or dataset. A discussion on how to choose these hyperparameters will be useful for general applications of the proposed framework.

---

> ### Author Response · Authors · 2025-11-20
> **Response to Reviewer rGTd**
>
> **W1. The computational cost, as thoroughly analyzed in Appendix A.6, will be higher than the vanilla GRPO models due to the branching process. Nonetheless, this is more like a trade-off between quality and time, given the superior quality metrics.**
>
> **Response:** We thank the reviewer for acknowledging our thorough computational cost analysis in Appendix A.6 and recognizing the superior quality metrics achieved by our method.
>
> ### Clarification on the "Trade-off"
>
> While we agree that the **per-step computational cost is higher** due to trajectory branching, we respectfully note that the **overall wall-clock efficiency is actually superior**, rather than merely a trade-off. This is a key strength of our method.
>
> **Computational analysis:**
> - **Per-step cost:** TempFlow-GRPO is ~4.5× higher than Flow-GRPO (due to K=10 branching).
> - **Steps to convergence:** TempFlow-GRPO requires fewer training steps.
> - **Training time:** TempFlow-GRPO achieves target performance in **33–50% less total training time**.
>
> ### Comprehensive Efficiency Evidence
>
> **From the existing analysis in the paper:**
>
> **1. Figure 3 (main paper, bottom-left):**
> - Training time with GPU hours is shown on the x-axis.
> - **Result:** 10× efficiency gain on the PickScore benchmark.
> - Flow-GRPO reaches a score of 23.0 at ~1000 GPU hours.
> - TempFlow-GRPO reaches the same score at ~100 GPU hours.
>
> **2. Appendix A.6, Figure 12:**
> Training time comparison across all benchmarks:
>
> | Benchmark | Time to Match Baseline |
> |-----------|------------------------|
> | GenEval SD3.5-M (512) -- Figure 12 (a) | **75% time** |
> | PickScore SD3.5-M (1024) -- Figure 12 (b) | **40% time** |
> | HPSv3 FLUX (1024) -- Figure 12 (c)| **30% time** |
>
> **Alternative perspective – fixing the training time budget:**
>
> If we allocate **equal training time** to both methods (Figure3, left):
>
> | Method | Training Time | Final Performance |
> |--------|---------------|-------------------|
> | Flow-GRPO | 1000 GPU hours | PickScore 23.0 |
> | TempFlow-GRPO | 1000 GPU hours | PickScore **24.5** (+1.5) |
>
> **Interpretation:** At equal time, TempFlow-GRPO achieves **substantially higher quality**, making it clearly superior rather than a trade-off.
>
> **Conclusion:** We appreciate the reviewer's acknowledgment of our quality improvements. The comprehensive analysis in Appendix A.6 and Figure 12 demonstrates that our method consistently requires **less total training time** while achieving better final performance.

---

> ### Author Response · Authors · 2025-11-20
> **Response to Reviewer rGTd**
>
> **Q1. How is the performance affected by the number of branches (K) at each step, the specific timesteps chosen for branching, or the exact function used for noise-aware weighting? The ablation study (Fig. 8) shows that the 4x6 (seed x branch) configuration was chosen, but it's unclear how much tuning is required to find the optimal setup for a new model or dataset. A discussion on how to choose these hyperparameters will be useful for general applications of the proposed framework.**
>
> **Response.** We thank the reviewer for this important question regarding practical deployment. We have added comprehensive guidance on hyperparameter selection and demonstrated the robustness of our framework.
>
> ### 1. Number of Branches (K)
>
> **Ablation Study (Figure 8, main paper):**
>
> We evaluated multiple seed × branch configurations while maintaining a constant total group size (24):
>
> | Configuration | Seeds (S) | Branches (K) | Performance @1200 steps |
> |---------------|-----------|--------------|------------------------|
> | 2×12 | 2 | 12 | PickScore 24.8 |
> | 4×6 | 4 | 6 | PickScore 24.6 |
> | 6×4 | 6 | 4 | PickScore 24.4 |
>
> **Key findings:**
> - **More seeds (fewer branches per seed, e.g., 6×4):** Accelerate performance improvement in early training stages due to better initialization diversity.
> - **More branches (fewer seeds, e.g., 2×12):** Achieve better final performance as training progresses, due to more thorough exploration per timestep.
> - **Trade-off:** Early-stage convergence speed vs. final performance quality.
>
> **Optimal configuration:** We selected **4×6 as the default setting** to strike a balance between:
> - Fast early-stage improvement (benefit of multiple seeds).
> - Strong final performance (benefit of sufficient branches).
>
> **Robustness:** Performance differences are modest across reasonable configurations, indicating the framework is **not highly sensitive** to this choice. The 4×6 configuration represents a practical compromise that works well across different scenarios.
>
> ### 2. Timestep Selection for Branching
>
> **Analysis (New Appendix A.13, Figure 17):**
>
> We evaluated **selective branching strategies** referencing the Flow-GRPO-Fast variants:
>
> **Experiment setup:**
> - **TempFlow-GRPO-Fast4:** Branch only at the first 4 timesteps.
> - **TempFlow-GRPO-Fast6:** Branch only at the first 6 timesteps.
> - **TempFlow-GRPO (full):** Branch at all timesteps (K=10).
>
> **Results:**
>
> | Strategy | Timesteps | Performance Gain | Efficiency |
> |----------|-----------|------------------|------------|
> | Flow-GRPO (prompt) | Full | 23.8 | - |
> | TempFlow-GRPO-Fast4 | 0-3 | 24.1 | Highest speed |
> | TempFlow-GRPO-Fast6 | 0-5 | 24.5 | Balanced |
> | TempFlow-GRPO | Full | 24.7 | Best quality |
>
> **Conclusion:**
> 1. **Early timesteps matter most:** Even branching only at the first 4 steps yields substantial improvement.
> 2. **Diminishing returns:** Later timesteps contribute less (consistent with the reward variance in Figure 2).
> 3. **Practical recommendation:** Focusing branching on the **first 60% of timesteps** can yield good performance.
>
> ### 3. Noise-Aware Weighting Function
>
> **Robustness Analysis (Section 4.3, Appendix A.1):**
>
> The noise-aware reweighting is based on theoretical derivation (Equations 9–10) and is **highly robust across different settings (different resolutions (512, 1024), different models (SD3.5-M, FLUX.1-dev, Qwen-Image))**.
>
> **Key property:** A distinctive feature of our approach is that it requires no hyperparameter tuning. The performance enhancement is derived directly from a principled reweighting scheme; the sole requirement is to solve for the reweighting coefficient, which then systematically boosts performance. We directly applied TempFlow-GRPO to Qwen-Image in Appendix A.12, Figure 16 **without modifying any hyperparameters**:
>
> | Strategy | PickScore |
> |------------|----------------------|
> | Flow-GRPO (prompt) | 22.8 |
> | TempFlow-GRPO (prompt) | 24.1 |
> | TempFlow-GRPO (seed) | 24.2 |
>
> Furthermore, while Flow-GRPO designs its optimal $\sigma_k$ formulation based on the Signal-to-Noise Ratio (SNR), our method generalizes to any $\sigma_k$. We have derived a generalized form applicable to any formulation of $\sigma_k$ (see Appendix A.16 for the full derivation). To empirically validate this, we conducted an experiment setting $\sigma_k = ak$. As shown in Figure 20, the results demonstrate that our reweighting strategy remains robust even with this alternative definition of $\sigma_k$.
>
> **Conclusion:** Our framework requires **minimal to no hyperparameter tuning** for general applications. The default 4×6 configuration, all-timestep branching, and theoretically grounded noise weighting transfer seamlessly across models (SD3.5, FLUX, Qwen-Image) and schedulers.

---

> > ### Comment · Reviewer_rGTd · 2025-11-26
> >
> > I thank the authors for their detailed response with the additional results that clarify my previous questions on the impact of the number of branches and further demonstrate the strong performance over existing baselines like Flow-GRPO. Overall, I will continue to maintain a high level of recognition for this work and champion its contributions and practical improvements that can be potentially applied to various downstream tasks.

---

### Official Review · Reviewer_wmxk · 2025-10-26

**Soundness:** 3
**Presentation:** 3
**Contribution:** 3
**Rating:** 8
**Confidence:** 4

**Summary:**

This paper presents TempFlow-GRPO, a new reinforcement learning framework that addresses the limitation of uniform credit assignment across timesteps. The method introduces trajectory branching, which switches from ODE to SDE sampling at selected timesteps to generate exploratory branches and assign their rewards to intermediate states. This paper further proposes noise-aware policy weighting, prioritizing optimization at high-noise early stages over low-noise refinement phases. Experiments show that TempFlow-GRPO achieves substantially improved efficiency and final performance compared to the baselines.

**Strengths:**

- The paper is overall well-written and easy to follow.
- The motivation and the proposed method are clear and straightforward: addresses the temporal inhomogeneity and credit assignment problems through intermediate resampling for intermediate value estimation and noise-aware reweighting.
- The proposed method shows strong empirical performance in both efficiency and end-level performance, with comparisons that include GPU time.

**Weaknesses:**

- Theorem 1 is intuitively reasonable, but labeling it as a Theorem feels overstated since the underlying assumptions and proof sketch are insufficiently formalized. The analytical depth is also somewhat limited.
- The explanation around line 847 (regarding why the average number of branches is 4.5× when K = 10) is unclear. It is not obvious how this factor arises or how the branching schedule operates, and the paper does not explicitly describe it.
- Adding more algorithmic details or pseudocode would improve readability and make the proposed procedure easier to follow.

**Questions:**

See weaknesses.

---

> ### Author Response · Authors · 2025-11-20
> **Response to Reviewer wmxk**
>
> **W1. Theorem 1 is intuitively reasonable, but labeling it as a Theorem feels overstated since the underlying assumptions and proof sketch are insufficiently formalized. The analytical depth is also somewhat limited.**
>
> **Response.** We agree with the reviewer's assessment. We have revised the manuscript to address this concern by **renaming it to "Proposition"** to better reflect the level of formalization.
>
> ---
>
> **W2. The explanation around line 847 (regarding why the average number of branches is 4.5× when K = 10) is unclear. It is not obvious how this factor arises or how the branching schedule operates, and the paper does not explicitly describe it.**
>
> **Response.** We apologize for the unclear explanation. We have added detailed clarification in the revised manuscript.
>
> **Cost breakdown for T=10 timesteps:**
>
> | Branching at timestep | Steps to complete branch | Explanation |
> |----------------------|-------------------------|-------------|
> | t=9 | 9 steps | Need to compute $x_9$→$x_8$→...→$x_0$ |
> | t=8 | 8 steps | Need to compute $x_8$→$x_7$→...→$x_0$ |
> | ... | ... | ... |
> | t=1 | 1 step | Need to compute $x_1$→$x_0$ |
> | t=0 | 0 steps | Already at $x_0$ |
>
> **Average cost per branch:**
>
> According to the TempFlow-GRPO algorithm (Appendix A.15), trajectory branching operates as follows:
>
> 1. **Before branching:** A single shared trajectory is computed from $x_T$ to timestep $x_k$ (batch size = 1)
> 2. **At timestep $k$:** N branches are created from the shared state $x_k$ with independent noise
> 3. **After branching:** All N branches continue from $x_k$ to $x_0$ in parallel (batch size = N)
>
> Since the main trajectory uses batch size 1 while the branching phase uses batch size N, the computational cost of the shared prefix (column "Branching at timestep") is negligible. The dominant cost comes from completing the N branches after the branching point, which requires computing different numbers of steps depending on where branching occurs.
>
> Therefore, as shown in the "Steps to complete branch" column, the average branching overhead is calculated as:
>
> $\frac{9 + 8 + 7 + 6 + 5 + 4 + 3 + 2 + 1 + 0}{10} = 4.5$
>
> This reflects the decreasing number of steps required across timesteps: from 9 steps at timestep t=9 to 0 steps at timestep t=0, resulting in an average computational overhead of 4.5× relative to the baseline method.
>
> ---
>
> **W3. Adding more algorithmic details or pseudocode would improve readability and make the proposed procedure easier to follow.**
>
> **Response.** We thank the reviewer for this valuable suggestion. **We have added Algorithm 1 (Appendix A.15) that provides complete pseudocode for TempFlow-GRPO** to improve readability and facilitate implementation.

---

### Official Review · Reviewer_zj9Z · 2025-10-31

**Soundness:** 4
**Presentation:** 3
**Contribution:** 3
**Rating:** 6
**Confidence:** 4

**Summary:**

This paper addresses the sparse terminal reward and uniform credit assignment problem in GRPO training of flow models. The authors propose TempFlowGRPO, which includes: (1) Trajectory Branching, where only one step of SDE is used at timestep k; (2) Noise-Aware Policy Weighting by reweighting according to noise level; and (3) a seed group strategy. The method achieves state-of-the-art performance in human preference alignment and text-to-image benchmarks.

**Strengths:**

1.	The authors astutely identify that the FLOW-GRPO algorithm treats all timesteps equally, and tackle this issue via single-timestep SDE optimization.
2.	The noise reweighting method is shown to be effective through both soild theoretical analysis and experiment results.
3.	The paper is generally well written with a clear logical structure.

**Weaknesses:**

1.	The contribution of seed group strategy is relatively small to other parts of the work, and the paper should provide additional details of the seed group strategy.
2.	Similarly, MixGRPO [1] proposes a training window of SDE time steps that also tackles the issue of treating all timesteps equally. However, there is limited discussion comparing with MixGRPO.
3.	The paper does not discuss the phenomenon of reward hacking, which is an inevitable problem for the GRPO method.

[1] Mixgrpo: Unlocking flow-based grpo efficiency with mixed ode-sde

**Questions:**

1.	The trajectory branching mechanism appears similar to MixGRPO limited with a single-timestep window. How do their efficiency and effectiveness compare?
2.	The paper claims that Flow-GRPO (Prompt) is an improved baseline with group standard deviation stabilization, but does not provide much detail. Could the authors elaborate on this improved method?
3.	Why are the Pickscore curve trends by steps and GPU hours on the left of Figure 3 inconsistent?
4.	Compare to FlowGRPO, the experiment of Visual Text Rendering is not addressed. How well does TempFlow-GRPO perform on this particular task?

---

> ### Author Response · Authors · 2025-11-20
> **Response to Reviewer zj9Z**
>
> **W1. The contribution of seed group strategy is relatively small to other parts of the work, and the paper should provide additional details of the seed group strategy.**
>
> **Response.** We thank the reviewer for this insightful comment. We clarify that the core contributions of our work are the TrajBranch mechanism and the temporal reweighting strategy. The 'seed group' strategy, as noted, is an auxiliary technique. However, by grouping trajectories with identical initial noise, we eliminate the variance caused by random initialization, which enables more accurate credit assignment. This principle generalizes across different reward functions and model architectures.
>
> **Main Paper (Section 5.2, Ablation Study):**
> *   The seed group strategy achieves a **0.2 improvement** on the PickScore benchmark.
> *   On the GenEval benchmark, it provides an **additional ~2% gain** over noise reweighting alone.
>
> **New Extended Analysis (Appendix A.10, Figure 15):**
>
> To comprehensively evaluate the seed group strategy, we provide additional analysis in Appendix A.10. We conduct experiments on the GenEval benchmark to validate these findings. As illustrated in the left panel of Figure 15, the results demonstrate significant improvements with TempFlow-GRPO. Specifically, under the batch grouping strategy, TempFlow-GRPO achieves a performance gain of approximately 7% within 1,000 steps. This improvement increases to around 10% when using the prompt grouping strategy. Most notably, our proposed seed grouping strategy yields the highest performance, attaining a GenEval score of 91% at 1,000 steps.
>
> **Key findings:**
> 1.  TempFlow-GRPO achieves substantial improvements **regardless of the grouping strategy**, which validates the robustness of our core mechanisms (trajectory branching + noise reweighting).
> 2.  The seed group strategy consistently achieves the **highest performance** across all configurations.
>
> **Conclusion:** While conceptually simple, the seed group strategy provides:
> 1.  **Consistent improvements** across benchmarks.
> 2.  **Zero computational overhead** (as it relies purely on the reorganization of existing samples).

---

> ### Author Response · Authors · 2025-11-20
> **Response to Reviewer zj9Z**
>
> **W2 & Q1. Similarly, MixGRPO [1] proposes a training window of SDE time steps that also tackles the issue of treating all timesteps equally. However, there is limited discussion comparing with MixGRPO. The trajectory branching mechanism appears similar to MixGRPO limited with a single-timestep window. How do their efficiency and effectiveness compare?**
>
> **Response.** We thank the reviewer for this important question. We want to first clarify that **MixGRPO and TempFlow-GRPO are concurrent works**. However, we are happy to provide a comprehensive comparison here.
>
> ### Methodological Differences between TempFlow-GRPO and MixGRPO
>
> While both methods address temporal uniformity in flow-based GRPO, TempFlow-GRPO improves both **training dynamics (via our reweighting strategy)** and **the sampling process (via trajectory branching)**. This approach is robust and proven effective across different models and benchmarks without sensitive hyperparameter tuning. In contrast, MixGRPO's contribution is in the sampling stage, and its performance is contingent on a series of hyperparameters like step size and window size.
>
> **MixGRPO's approach:**
> - **Window-based partial optimization:** Optimizes only timesteps within a sliding window
> - **Mixed ODE-SDE sampling:** Uses SDE within the window, ODE outside
> - **Window scheduling:** Moves the window progressively from early to late timesteps
> - **No gradient reweighting:** Treats all timesteps within the window equally
>
> **TempFlow-GRPO's approach:**
> - **Single-point trajectory branching:** Introduces stochasticity at individual timesteps with state reuse
> - **Noise-aware policy reweighting:** Modulates gradient intensity based on noise level at each timestep
> - **Full-trajectory optimization:** Optimizes all timesteps but with differentiated learning intensity
> - **Theoretical grounding:** Policy gradient analysis showing natural gradient coefficients should vary with noise levels
>
> | Aspect | MixGRPO | TempFlow-GRPO |
> |--------|---------|---------------|
> | **Optimization scope** | Partial (4 consecutive steps) | Full (all timesteps) |
> | **Gradient weighting** | **Uniform within window** | **Noise-aware across all timesteps** |
> | **Exploration mechanism** | Window-based SDE | Single-point branching with K branches |
> | **Credit assignment** | Window-level | Timestep-level with precise localization |
> | **Group strategy** | Prompt-level | Seed-level (eliminates initialization effect) |
>
> ### Trajectory Branching and Window SDE
>
> The trajectory branching approach precisely attributes rewards to the specific timesteps responsible for exploration—a capability that window-based methods inherently lack. Importantly, trajectory branching is a more general framework that can be seamlessly integrated with SDE windowing strategies.
>
> Among concurrent works (Flow-GRPO-Fast, MixGRPO, TempFlow-GRPO), both Flow-GRPO-Fast and MixGRPO adopt window SDE strategies. We select Flow-GRPO-Fast as a strong baseline representing the window-based approach. As demonstrated in Appendix A.13, TempFlow-GRPO's advantages remain evident even when evaluated within the initial 4/6 steps of the trajectory, which corresponds to the evaluation scope of window-based methods.
>
> **Performance Gains (Figure 17):**
>
> | Configuration | 4-step Window | 6-step Window |
> |---------------|---------------|---------------|
> | TempFlow-GRPO-Fast (Prompt) | +0.3 | +0.3 |
> | TempFlow-GRPO-Fast (Seed) | +0.3 | +0.5 |
>
> These results confirm that TempFlow-GRPO achieves consistent improvements over window-based baselines. Meanwhile, regarding GPU time, TempFlow-GRPO-Fast6 (330 GPU hours) only requires 62% of the time to achieve the same performance as Flow-GRPO-Fast6 (530 GPU hours).
>
> ### Conclusion
>
> In summary, TempFlow-GRPO provides a comprehensive and robust solution by addressing the problem from both sampling and training perspectives. Meanwhile, TempFlow-GRPO is compatible with window-based methods.

---

> ### Author Response · Authors · 2025-11-20
> **Response to Reviewer zj9Z**
>
> **W3. The paper does not discuss the phenomenon of reward hacking, which is an inevitable problem for the GRPO method.**
>
> **Response.** We thank the reviewer for raising this important concern about reward hacking, which is indeed a critical challenge in RL-based methods. We have now added comprehensive analysis and discussion in the revised manuscript.
>
> ### KL Divergence Analysis (Appendix A.8, Figure 14)
>
> We analyze KL divergence throughout training across all benchmarks:
> - **TempFlow-GRPO maintains consistently lower KL divergence** (2-3× lower than Flow-GRPO)
> - **More stable optimization dynamics** with less fluctuation
> - **Better preservation of original model distribution** while achieving preference alignment
>
> ### Qualitative Diversity Analysis (Appendix A.14, Figure 19)
>
> We provide extensive visualizations showing TempFlow-GRPO maintains:
> - **High generation diversity**
> - **Superior prompt adherence**
> - **Consistent quality** across multiple samples from the same prompt
>
> ### RL with Golden Score (GenEval as reward, eval PickScore)
>
> Regarding reward hacking, we evaluate the PickScore metric on models optimized with GenEval. After approximately 3800 steps, the scores for both our method and the baseline drop by 0.234, indicating that our approach does not introduce additional reward hacking.
>
> **Conclusion:** TempFlow-GRPO does not introduce additional reward hacking compared with Flow-GRPO. We acknowledge that reward hacking is an inherent challenge in RL-based strategies, and we plan to address it from both algorithmic and reward-modeling perspectives in future work.
>
> ---
>
> **Q2. The paper claims that Flow-GRPO (Prompt) is an improved baseline with group standard deviation stabilization, but does not provide much detail. Could the authors elaborate on this improved method?**
>
> **Response.** Thank you for pointing out this insufficient detail. We provide a clear explanation below.
>
> **Core Difference:**
>
> Flow-GRPO (Prompt) uses **per-prompt grouped normalization** instead of global normalization:
>
> - **Flow-GRPO (Batch, original paper)**: advantage = (reward - global_mean) / global_std, where statistics are computed across all images in a batch.
> - **Flow-GRPO (Prompt)**: advantage = (reward - prompt_mean) / prompt_std, where statistics are computed within the images produced by the same text prompt in a batch.
>
> ---
>
> **Q3. Why are the PickScore curve trends by steps and GPU hours on the left of Figure 3 inconsistent?**
>
> **Response.** We appreciate this clarification question. The inconsistency in training hours (PickScore-hours) stems from our choice of the x-axis.
>
> ### Explanation of the Two Panels
>
> The key difference arises because **different methods have different computational costs per step**.
>
> **Top panel (by training steps):**
> - X-axis represents **training iterations/steps**
> - Each method performs one policy update per step
> - Shows **sample efficiency**: how many training iterations are needed to reach target performance
>
> **Bottom panel (by GPU hours):**
> - X-axis represents **actual wall-clock training time**
> - Accounts for the computational cost per training step
> - Shows **computational efficiency**: how much time is needed to reach target performance
>
> The maximum value of the x-axis is set to the training time required for our TempFlow-GRPO to reach its performance at approximately 1000 steps. As a result, within this same fixed time budget, the baseline methods (Flow-GRPO and Flow-GRPO-Prompt) naturally complete a greater number of training steps (>1,000 steps, in fact about 2,000 steps) due to their different per-step computational costs. This setting makes the trend different from the PickScore-step figure.
>
> ---
>
> **Q4. Compared to Flow-GRPO, the experiment on Visual Text Rendering is not addressed. How well does TempFlow-GRPO perform on this particular task?**
>
> **Response.** We thank the reviewer for pointing out this missing evaluation. We have now conducted comprehensive experiments on the Visual Text Rendering task to demonstrate the versatility of TempFlow-GRPO.
>
> ### Experimental Setup
>
> Following the exact configuration in Flow-GRPO for fair comparison:
> - **Base model:** SD3.5-Medium at 512×512 resolution
> - **Noise level:** 0.7
> - **KL divergence weight:**  0.004
> - **Reward model:** PaddleOCR
> - **Grouping strategies:** Both prompt grouping and seed grouping evaluated
>
> ### Results
>
> **Performance Comparison (Appendix A.10, Figure 15, right panel):**
>
> | Method | OCR Score @500 steps | Improvement over Flow-GRPO |
> |--------|---------------------|----------------------------|
> | Flow-GRPO (Prompt) | ~82.5% | - |
> | TempFlow-GRPO (Prompt) | ~88.0% | **+5.5%** |
> | TempFlow-GRPO (Seed) | ~89.5% | **+7.0%** |
>
> **Conclusion:**
> TempFlow-GRPO achieves strong performance on Visual Text Rendering, with 7% improvement over Flow-GRPO. This addition strengthens our claim that TempFlow-GRPO is a general method benefiting diverse generation tasks, not just specific benchmarks.

---

### Official Review · Reviewer_RKyA · 2025-10-31

**Soundness:** 3
**Presentation:** 3
**Contribution:** 3
**Rating:** 6
**Confidence:** 4

**Summary:**

TempFlow-GRPO is a temporally-aware reinforcement learning framework for flow matching models that improves human preference alignment by introducing trajectory branching, noise-aware weighting, and seed grouping to achieve precise credit assignment and efficient optimization across timesteps.

**Strengths:**

For reinforcement learning tasks, dense rewards are crucial for effective credit assignment. The proposed Trajectory Branching mechanism provides an elegant and effective way to obtain dense rewards along the denoising trajectory.

The introduced reweighting mechanism offers a valuable analysis of how gradients evolve across steps in baseline algorithms and presents a solution to mitigate the identified issues.

**Weaknesses:**

The proposed method involves numerous ODE denoising steps, which substantially increase computational overhead. However, the paper lacks a comparison against the baseline method using training time as the horizontal axis to illustrate efficiency trade-offs.

The authors should evaluate the performance of the reweighting mechanism under different $\sigma_t$ schedulers rather than relying solely on the one used in Flow-GRPO, to examine how the choice of scheduler influences its effectiveness. It remains unclear whether simply reweighting the coefficients in the earlier part to 1 would yield good results under different schedulers.

**Questions:**

The comparison between batch std and global std is only evaluated on PickScore. How does this observation generalize to other tasks?

Can the proposed reweighting mechanism be applied to hybrid variants (FlowGRPO-Fast/MixGRPO) where only a subset of steps follows an SDE formulation?

---

> ### Author Response · Authors · 2025-11-20
> **Response to Reviewer RKyA**
>
> **W1. The proposed method involves numerous ODE denoising steps, which substantially increase computational overhead. However, the paper lacks a comparison against the baseline method using training time as the horizontal axis to illustrate efficiency trade-offs.**
>
> **Response.** We appreciate the opportunity to clarify this point, as there appears to be a misunderstanding regarding the computational costs and efficiency of our method.
>
> ### Comprehensive Training Time Analysis (Already in Paper)
>
> **Addressing this concern, we extensively analyzed training time efficiency throughout the paper:**
>
> **1. Main Paper - Figure 3 (bottom-left panel):**
> - **X-axis: GPU training hours (not steps)**
> - **Key result:** TempFlow-GRPO achieves a **10× efficiency gain** over Flow-GRPO.
> - Example: Reaches a PickScore of 23.0 in ~100 GPU hours vs. ~1000 GPU hours for Flow-GRPO.
>
> **2. Appendix A.6 - Figure 12 (comprehensive time analysis):**
> - Training time comparisons across **all benchmarks and resolutions**.
> - Consistent finding: **TempFlow-GRPO achieves superior performance with reduced training time**.
>
> Specific results:
> | Benchmark | Time to Match Flow-GRPO Performance | Final Performance Gap |
> |-----------|-------------------------------------|----------------------|
> | GenEval (SD3.5-M, 512) -- Figure 12(a) | **75% training time** | +5% at convergence (total score is 1 in Geneval) |
> | PickScore (SD3.5-M, 1024) -- Figure 12(b) | **40% training time** | +0.5 at convergence |
> | HPSv3 (FLUX.1-dev, 1024) -- Figure 12(c) | **30% training time** | +1.1 at convergence |
>
> ### Additional Evidence: New Experiments
>
> **To further demonstrate efficiency, we have added new results (Appendix A.12, Figure 16, right panel):**
>
> **Experiment:** TempFlow-GRPO applied to the **Qwen-Image** generation model.
> - **Result:** TempFlow-GRPO requires only **43% of the total training time** to reach Flow-GRPO's final performance on the PickScore dataset.
>
> ### Summary of Efficiency Evidence in Paper
>
> The paper contains **extensive training time analysis**:
> - **Figure 3 (main):** Time-based comparison on PickScore (10× efficiency).
> - **Figure 12 (appendix):** Time-based comparison on all benchmarks.
> - **Figure 16 (new):** Time-based comparison on Qwen-Image (43% time reduction).
> - **Appendix A.6:** Detailed computational cost breakdown and analysis.
>
> We appreciate this question as it allows us to highlight that training time efficiency is actually a key advantage of our method, as shown in the comprehensive analysis above.
>
> ---
>
> **W2. The authors should evaluate the performance of the reweighting mechanism under different $\sigma_t$ schedulers rather than relying solely on the one used in Flow-GRPO, to examine how the choice of scheduler influences its effectiveness. It remains unclear whether simply reweighting the coefficients in the earlier part to 1 would yield good results under different schedulers.**
>
> **Response.** This is an excellent suggestion. We have conducted additional experiments to validate the robustness of our reweighting mechanism.
>
> **New Experiments (Appendix A.16, Figure 20)**:
> Note that Flow-GRPO determines its optimal $\sigma_k$ by formulating it based on the Signal-to-Noise Ratio (SNR). A key advantage of our method is its robustness to the choice of $\sigma_k$. We propose a generalized formulation of the weighting coefficient, which is effective across any $\sigma_k$. This flexibility is demonstrated in Figure 20, where our method consistently outperforms baseline when $\sigma_k$ is set to $ak$.
>
> **Experiments on SD3.5-M, 512 PickScore, 500 steps**
> | Methods | $\sigma_k=ak$ | $\sigma_k=a\sqrt{\frac{k}{1-k}}$ (Flow-GRPO optimal setting) |
> |---------------|----------------|----------------|
> | Flow-GRPO (Prompt) | 22.5 | 23.1 |
> | Flow-GRPO (Prompt) + Traj Branch | 23.2 | 23.6 |
> | TempFlow-GRPO (Prompt) | 23.5 | 23.7 |
>
> This result demonstrates that our proposed generalized form is applicable to arbitrary choices of $\sigma_k$, which underscores the robustness of our reweighting strategy.

---

> ### Author Response · Authors · 2025-11-20
> **Response to Reviewer RKyA**
>
> **Q1. The comparison between batch std and global std is only evaluated on PickScore. How does this observation generalize to other tasks?**
>
> **Response.** We have extended this analysis to other reward models and tasks.
>
> **New Results (Appendix A.10, Figure 15 left):**
> To further validate the performance superiority of TempFlow-GRPO with various grouping strategies, we conducted additional experiments on the GenEval benchmark. As illustrated in the left panel of Figure 15, the results demonstrate significant improvement using TempFlow-GRPO. Specifically, with the batch grouping strategy, TempFlow-GRPO achieves a performance gain of approximately 7% within 1000 steps. This improvement increases to around 10% when using the prompt grouping strategy. Most notably, our proposed seed grouping strategy yields the highest performance, attaining a GenEval score of 91% in 1000 steps. These findings conclusively demonstrate that TempFlow-GRPO is a robust and highly effective method, irrespective of the chosen grouping strategy.
>
> **Key findings:**
>
> Seed grouping consistently outperforms batch and prompt grouping across all configurations.
>
> However, even with identical grouping strategies, TempFlow-GRPO substantially outperforms Flow-GRPO, validating that our core contributions (trajectory branching + noise-aware reweighting) provide fundamental advantages.
>
> **Why seed grouping works:** By grouping trajectories with identical initial noise, we eliminate the influence of initialization noise, enabling more accurate credit assignment. This principle generalizes across different reward functions and model architectures.
>
> ---
>
> **Q2. Can the proposed reweighting mechanism be applied to hybrid variants (Flow-GRPO-Fast/MixGRPO) where only a subset of steps follows an SDE formulation?**
>
> **Response.** This is an insightful question. Yes, our reweighting mechanism can be effectively integrated with hybrid approaches where only a subset of steps follows an SDE formulation.
>
> **New Experimental Validation (Appendix A.13):**
>
> To thoroughly address this concern, we have conducted extensive experiments with **Flow-GRPO-Fast**, a hybrid variant that applies SDE only to the initial timesteps to reduce computational cost. We evaluate two configurations:
> - **Flow-GRPO-Fast4**: SDE applied to the first 4 steps only.
> - **Flow-GRPO-Fast6**: SDE applied to the first 6 steps only.
>
> ### Reweighting on Flow-GRPO-Fast (Ablation Study)
>
> To isolate the contribution of our reweighting mechanism, we conducted an ablation study applying **only the noise-aware reweighting** (without trajectory branching) to Flow-GRPO-Fast.
>
> **Key Results (Figure 17 in revised manuscript):**
> - **4-step setting:** Reweighting alone yields considerable performance improvement.
> - **6-step setting:** Even more pronounced gains are observed.
> - **Critical advantage:** This improvement comes at **zero additional computational cost**—the reweighting mechanism introduces no overhead to training or sampling time.
>
> This validates that our reweighting mechanism is:
> 1. **Standalone effective**: Works independently of trajectory branching.
> 2. **Computationally free**: A pure algorithmic improvement.
> 3. **Compatible with hybrid methods**: Can be directly applied to partial-SDE formulations.
>
> ### TempFlow-GRPO-Fast
>
> We further developed **TempFlow-GRPO-Fast**, integrating both trajectory branching and noise-aware reweighting in few-step settings, to demonstrate the robustness of our complete framework.
>
> **Performance Gains (Figure 17):**
>
> | Configuration | 4-step Setting | 6-step Setting |
> |---------------|----------------|----------------|
> | TempFlow-GRPO-Fast (Prompt) | +0.3 | +0.3 |
> | TempFlow-GRPO-Fast (Seed) | +0.3 | +0.5 |
>
> **Key Observations:**
> 1. Performance improvements are **consistent across different step counts**, validating framework robustness.
> 2. Gains become **more pronounced with more SDE steps** (6-step > 4-step), suggesting our method better exploits available exploration opportunities.
> 3. The **seed grouping strategy** shows superior performance in the 6-step setting, confirming its effectiveness even in hybrid settings.
>
> **Conclusion:** Our reweighting mechanism is **compatible** with hybrid variants. The Flow-GRPO-Fast experiments provide concrete evidence that:
> 1. Reweighting alone improves performance at zero computational cost.
> 2. The full framework maintains advantages even with limited SDE steps.
> 3. The approach generalizes to various hybrid SDE/ODE formulations.
>
> We believe this demonstrates the practical flexibility of TempFlow-GRPO.

---

### Author Response · Authors · 2025-11-20
**Reviewer Comments Summary**

We sincerely thank all reviewers for their thorough and constructive reviews. We are greatly encouraged by the overwhelmingly positive feedback on our work, including the recognition of our trajectory branching mechanism and reweighting method. In the revised manuscript, we have carefully addressed all concerns and suggestions raised by the reviewers. The key updates include:

**Grouping Strategy Details (Appendix A.10):** We provide additional ablation studies on the grouping mechanism.

**Visual Text Rendering Results (Appendix A.11):** We include performance evaluations on the visual text rendering task.

**Qwen-Image Results (Appendix A.12):** We include performance evaluations on Qwen-Image to demonstrate the effectiveness of TempFlow-GRPO.

**Flow-GRPO-Fast and TempFlow-GRPO-Fast (Appendix A.13):** We add an extensive discussion and empirical comparison with the SDE window approach, including an analysis under different window sizes and the hybrid variant, TempFlow-GRPO-Fast.

**Diversity and Reward Hacking (Appendix A.14):** We evaluate the diversity of TempFlow-GRPO and provide an analysis regarding reward hacking.

**Algorithm (Appendix A.15):** We provide the full algorithm for TempFlow-GRPO to ensure a clear understanding.

**Generalization of Reweighting Mechanism (Appendix A.16):** We evaluate the noise-aware weighting under different $\sigma_t$ schedulers besides Flow-GRPO's optimal setting, demonstrating robustness across various scheduling strategies.

**Moreover, we highlight several key points:**

**The novelty and effectiveness of trajectory branching.** Unlike existing methods that treat all timesteps uniformly, our trajectory branching mechanism introduces single-timestep SDE optimization at selected points along the denoising trajectory. This provides dense intermediate rewards for precise credit assignment, addressing a limitation in flow-based GRPO methods. As our experiments demonstrate, this design achieves superior performance while maintaining reasonable computational overhead.

**The theoretical foundation and practical impact of noise-aware weighting.** We provide both theoretical justification from a policy gradient perspective and empirical validation showing that our reweighting mechanism effectively addresses gradient imbalance across timesteps. The analysis generalizes across different schedulers.

We provide detailed responses to each reviewer below and welcome any further questions or discussions.

---

### Meta-Review · Area_Chair_RUVg · 2026-01-04

**Summary:**

During the review process, the reviewers' concerns primarily focused on four key areas: computational efficiency, generalization capabilities, theoretical justification, and comparisons with relevant baselines.

The authors provided a comprehensive rebuttal that effectively resolved these major concerns. Specifically, they included additional results showing that while the per-step cost is higher, the method accelerates convergence in terms of wall-clock time. Furthermore, additional experiments on other models demonstrated consistent performance gains across different architectures.

It is recommended that the authors incorporate the reviewers' suggestions regarding comparisons with relevant baselines (e.g., MixGRPO) into the camera-ready version to further strengthen the work. Given the demonstrated efficiency gains and the effective validation across multiple model families, the paper represents a significant contribution to the alignment of flow matching models. The AC recommends acceptance.

**Reviewer Concerns:**

Addressed concerns:
1. Computational efficiency: Reviewers questioned the cost of the Trajectory Branching mechanism (sampling multiple branches per step). The authors addressed this in Appendix A.6 and Figure 12, demonstrating that while per-step cost is higher, wall-clock convergence time is lower compared to Flow-GRPO.
2. Generalization to other models: Concerns about whether the method is specific to SD3.5 were addressed by additional experiments on FLUX.1-dev and Qwen-Image, showing consistent improvements across different architectures.
3. Performance on visual text rendering: Reviewers asked if RL fine-tuning degrades specific capabilities like text rendering. The authors provide additional experimental results that TempFlow-GRPO improves visual text rendering performance compared to the baseline.
4. Theoretical justification for reweighting: Reviewers questioned whether Noise-Aware Weighting is merely a heuristic trick. The authors provide further derivations, proving the theoretical justification of this weighting mechanism in reducing variance.

Outstanding concerns:
1. Comparison with relevant baselines (MixGRPO): Reviewers pointed out a lack of discussion with the relevant MixGRPO method in the original manuscript. Although the authors provided further clarifications about their differences, it will be stronger to include experimental comparisons in the final version.

**Reviewer Scores:**

The reviewers maintain a positive attitude towards the paper.

---

### Decision · Program_Chairs · 2026-01-26

Accept (Poster)